# Pixel Reasoner: Incentivizing Pixel-Space Reasoning with Curiosity-Driven Reinforcement Learning

**Alex Su**♠♡∗ **Haozhe Wang**♢♡∗ **Weiming Ren**♡†, **Fangzhen Lin**♢, **Wenhu Chen**♡†

University of Waterloo♡, HKUST♢, USTC♠, Vector Institute†

Project Page: `https://tiger-ai-lab.github.io/Pixel-Reasoner/`

## Abstract

Chain-of-thought reasoning has significantly improved the performance of Large Language Models (LLMs) across various domains. However, this reasoning process has been confined exclusively to textual space, limiting its effectiveness in visually intensive tasks. To address this limitation, we introduce the concept of reasoning in the pixel-space. Within this novel framework, Vision-Language Models (VLMs) are equipped with a suite of visual reasoning operations, such as zoom-in and select-frame. These operations enable VLMs to directly inspect, interrogate, and infer from visual evidences, thereby enhancing reasoning fidelity for visual tasks. Cultivating such pixel-space reasoning capabilities in VLMs presents notable challenges, including the model's initially imbalanced competence and its reluctance to adopt the newly introduced pixel-space operations. We address these challenges through a two-phase training approach. The first phase employs instruction tuning on synthesized reasoning traces to familiarize the model with the novel visual operations. Following this, a reinforcement learning (RL) phase leverages a curiosity-driven reward scheme to balance exploration between pixel-space reasoning and textual reasoning. With these visual operations, VLMs can interact with complex visual inputs, such as information-rich images or videos to proactively gather necessary information. We demonstrate that this approach significantly improves VLM performance across diverse visual reasoning benchmarks. Our 7B model, `Pixel-Reasoner`, achieves 84% on V* bench, 74% on TallyQA-Complex, and 84% on InfographicsVQA, marking the highest accuracy achieved by any open-source model to date. These results highlight the importance of pixel-space reasoning and the effectiveness of our framework.

## 1 Introduction

Recent advancements have demonstrated remarkable progress in developing complex reasoning abilities in Vision-Language Models (VLMs). Leading models, such as OpenAI GPT4o/GPT-o1 [Hurst et al., 2024, Jaech et al., 2024a], Gemini-2.5 [Team et al., 2023], VL-Rethinker [Wang et al., 2025] achieve superior performance on various multimodal reasoning benchmarks like MathVista [Lu et al., 2023], MMMU [Yue et al., 2024], MEGA-Bench [Chen et al., 2024], etc. A common paradigm underpinning these state-of-the-art VLMs involves processing multimodal queries to extract relevant cues, followed by a reasoning process (CoT [Wei et al., 2022]) conducted purely in the textual format.

Despite their success, the prevailing textual reasoning paradigm faces an inherent limitation: relying solely on text tokens to express intermediate reasoning steps can constrain the depth and accuracy achievable by Vision-Language Models (VLMs) on visually intensive tasks. The lack of direct interaction with visual inputs—such as drawing lines/marks, highlighting regions, or zooming

---

∗These authors contributed equally and are listed alphabetically. Haozhe as Project Lead.
Corresponding to: dlwlrma314516@gmail.com, jasper.whz@outlook.com, wenhuchen@uwaterloo.ca

in—hinders the model's ability to interact with the information-rich images. As a result, VLMs often struggle to capture fine-grained visual details, including tiny objects, subtle spatial relationships, small embedded text, and nuanced actions in videos.

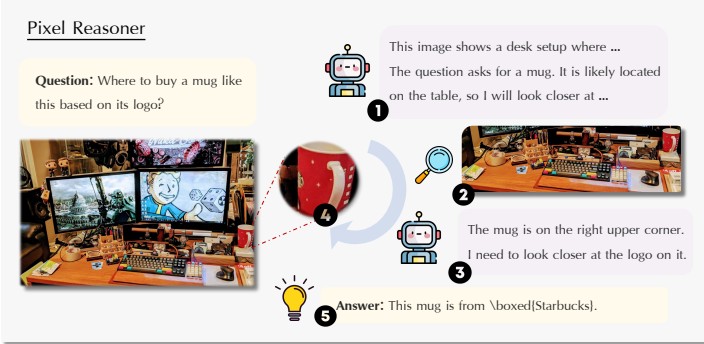

Figure 1: **Illustration of Pixel Reasoner.** When asked a visually-rich question, `Pixel-Reasoner` first inspects the visual inputs. Then it iteratively refines its understanding and evolve its reasoning by leveraging visual operations, such as ZOOM-IN for images and SELECT-FRAMES for videos, ultimately arriving at a conclusion.

Figure 2: The Learning Trap. Our approach combines a warm-start instruction tuning phase and curiosity-driven RL phase to overcome the learning trap.

These limitations motivates a fundamental rethinking of how VLMs engage with the visual modality during reasoning more seamlessly. This leads us to pose the research question:

**Can VLMs perform reasoning steps more directly within the visual modality itself, leveraging computational visual manipulations as actions to guide reasoning?**

We introduce the concept of pixel-space reasoning, proposing a novel paradigm where reasoning is not exclusively confined to verbalized format but actively incorporates operations applied directly to the visual inputs. Rather than solely translating visual observations into textual cues, the model could actively manipulate and interact with the visual information throughout the reasoning process – employing operations like 'ZOOM-IN', or 'SELECT-FRAME'. These visual operations serve as integral steps within its reasoning chain, empowering the model to inspect, interrogate, and infer from visual evidence with enhanced fidelity. We frame this problem as developing a VLM endowed with a suite of visual operations. This novel framework involves strategically selecting and applying appropriate visual operations to the visual inputs, progressively refining its understanding, evolving its reasoning and ultimately arriving at a conclusion. To instill this novel capability of pixel-space reasoning, we follow the common post-training paradigm of instruction tuning and reinforcement learning (RL). However, cultivating pixel-space reasoning presents significant challenges.

Firstly, existing VLMs exhibit limited zero-shot proficiency in executing pre-defined visual operations, thus requiring meticulous instruction tuning to establish a foundational understanding of these new visual operations. This initial training phase must also preserve the model's inherent self-correction abilities, thereby preparing for trial-and-errors in the subsequent RL phase.

Secondly, the warm-started model exhibits a significant disparity in proficiency between its well-established textual reasoning and its emergent pixel-space reasoning capabilities, which creates a "learning trap" that impedes the effective acquisition of pixel-space reasoning. On one hand, the model's initial incompetence in visual operations garners more negative feedback than textual reasoning. On the other hand, a significant portion of training queries may not strictly necessitate visual operations, allowing the model to bypass these under-developed skills. These factors trap the cultivation of pixel-space reasoning, causing the premature cessation of efforts to utilize visual operations and improve pixel-space reasoning.

To address these challenges, our approach combines a warm-start instruction tuning phase and a reinforcement learning phase. For instruction tuning, we synthesize 7,500 reasoning traces that facilitate the cultivation of both mastery over visual operations and self-correction capabilities. Following this meticulous warm-start instruction tuning, our RL approach leverages a curiosity-driven reward scheme to balance the exploration and exploitation of pixel-space reasoning to incentivize the pixel-level reasoning. The RL phase collect another 7,500 examples from several public image and video datasets [Feng et al., 2025, Xu et al., 2025]. Our final model `Pixel-Reasoner`, built on

top of Qwen2.5-VL-7B [Bai et al., 2025], is able to show significant improvement across several visual reasoning benchmarks (with information-rich images/video) like V* [Wu and Xie, 2024], TallyQA [Acharya et al., 2019], MVBench [Li et al., 2024a] and InfographicsVQA [Mathew et al., 2021]. On these benchmarks, `Pixel-Reasoner` shows best known open-source performance and even exceeds proprietary models like Gemini-2.5-Pro [Team et al., 2024a] and GPT-4o [Hurst et al., 2024]. We further conduct comprehensive ablation studies to provide insights into how our framework effectively cultivates pixel-space reasoning. Our contributions are listed as follows:

1. We introduce the concept of *pixel-space reasoning* for the first time.
2. We identified a learning trap when cultivating this novel reasoning ability.
3. We proposed a novel two-staged post-training approach, featuring a meticulous instruction tuning stage, and a curiosity-driven RL stage.
4. We achieved state-of-the-art results on visually-intensive benchmarks with pixel-space reasoning.

## 2 Problem Formulation

We introduce *pixel-space reasoning*, a novel paradigm enabling models to integrate operations directly applied to visual inputs, rather than solely relying on textual reasoning. Formally, consider a vision-language query $\mathbf{x} = [V, L]$, where V represents visual inputs (e.g., images or videos) and L is the textual query. A model $\pi_\theta$ constructs a solution $\mathbf{y} = [y_1, \ldots, y_n]$ via an iterative reasoning process in both pixel and textual space. At each step t, the model generates a reasoning segment $y_t \sim \pi_\theta(\cdot|\mathbf{x}, \mathbf{y}_{t-1})$ conditioned on the initial query $\mathbf{x}$ and the set of all preceding reasoning steps $\mathbf{y}_{t-1} = [y_1, \ldots, y_{t-1}]$. Unlike the predominant textual reasoning paradigm, pixel-space reasoning allows each reasoning step $y_t$ to be one of two types:

- **Textual Thinking**: Steps that involve reasoning purely within the textual domain, like calculating an equation or use domain knowledge to derive a conclusion, etc.
- **Visual Operations**: Steps that activate visual operations to directly manipulate or extract information from the visual inputs. A visual operation $y_t$ involves invoking a predefined function $f$, yielding an execution outcome $\mathbf{e}_t = f(y_t)$. For instance, a model might generate $y_t$ to trigger a `select_frame` operation, $f_{\text{SF}}$, with specified arguments (e.g., "target_frame") in $y_t$ to retrieve visual tokens $\mathbf{e}_t$ for a particular frame. The reasoning step is then updated to $y_t \leftarrow \text{concat}(y_t, \mathbf{e}_t)$, incorporating the execution outcome $\mathbf{e}_t$ for subsequent reasoning.

This iterative reasoning process concludes when a designated end token is generated. We aim to cultivate *pixel-space reasoning* via reinforcement learning (RL), where the objective is to optimize a Vision-Language Model (VLM) policy $\pi_\theta$ that maximizes the expected reward over a dataset $\mathcal{D}$:

$$\max_\theta \mathbb{E}\left[r(\mathbf{x}, \mathbf{y})\Big|\mathbf{x} \sim \mathcal{D}, \mathbf{y} \sim \pi_\theta(\mathbf{y}|\mathbf{x})\right]$$

A common approach for $r(\mathbf{x}, \mathbf{y})$ is to adopt a binary correctness reward, which assesses the correctness of the generated solution $\mathbf{y}$ for a given query $\mathbf{x}$ [DeepSeek-AI et al., 2025, Liu et al., 2025]:

$$r(\mathbf{x}, \mathbf{y}) = \begin{cases} 1 & \text{if the solution } \mathbf{y} \text{ contains the correct answer to query } \mathbf{x}, \\ 0 & \text{otherwise.} \end{cases}$$

In this work, we focus on two types of visual inputs: images and videos. We specifically consider two types of visual operations: ZOOM-IN for inspecting details within a specified region of a target image, and SELECT-FRAME for analyzing specific frames in a video sequence. Detailed protocols for these visual operations are provided in the appendix.

## 3 Warm-Start Instruction Tuning

We aim to cultivate a novel pixel-space reasoning paradigm leveraging existing Visual-Language Models (VLMs). However, instruction-tuned models such as Qwen2.5-VL-Instruct exhibit limited zero-shot proficiency in executing novel visual operations (as shown in the analysis in the appendix), likely due to their absence in standard training data. To lay the groundwork for utilizing visual operations in subsequent reinforcement learning, we describe in this section our approach to data curation and instruction tuning.

**Collect Seed Datasets.** Our data curation pipeline is designed to collect high-quality pixel-space reasoning trajectories. These trajectories are intended to serve as expert demonstrations for our policy,

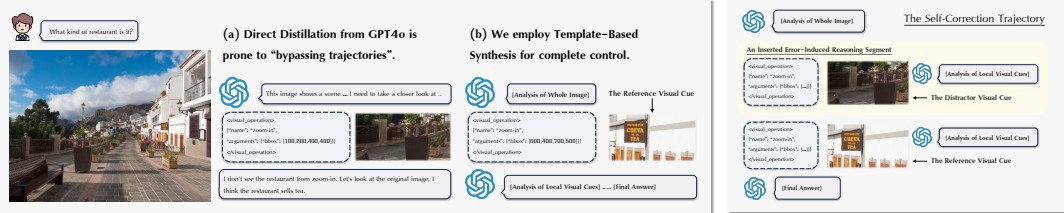

Figure 3: Direct Distillation from GPT4o may generate "bypassing trajectories" where the model ignores the visual operations and performs textual reasoning. We thus adopt a template-based synthesis strategy.

Figure 4: We synthesize self-correction trajectories by inserting erroneous reasoning segments.

showcasing the effective utilization of visual operations. Therefore, we first select three datasets: SA1B [Kirillov et al., 2023], FineWeb [Ma et al., 2024] and STARQA [Wu et al., 2024]. These datasets offer diverse modalities and contents, spanning natural scenes with extensive segmentation masks, diverse web-pages, and real-world videos requiring situated reasoning. Across all three datasets, their high visual complexity provides rich visual information for fine-grained analysis, and their explicit annotations serve as crucial reference visual cues for trajectory synthesis. A detailed description of these datasets can be found in the appendix.

**Localize Reference Visual Cues.** To ensure that the visual operations are genuinely necessary for resolving the vision-language queries, we selected or synthesized queries that specifically require the localization of fine-grained visual cues within the rich visual information. The FineWeb and STARQA datasets already provide vision-language queries paired with reference visual cues for answers. For the SA1B dataset, we first leveraged GPT-4o to identify specific target visual details within an image, such as small objects or particular attributes. Subsequently, we prompted GPT-4o to generate a natural language query based on the identified detail and the corresponding image, formulating a fine-grained visual question that necessitates locating that specific cue.

**Synthesize Expert Trajectories.** Based on the curated vision-language queries requiring fine-grained visual analysis, we then synthesize expert trajectories using GPT-4o. As illustrated in Fig. 3 (a), we observed that direct distillation from GPT-4o sometimes resulted in "bypassing trajectories". In these cases, GPT-4o could occasionally bypass errorroneous visual operations and arrive at the correct final answer solely through its textual reasoning capabilities. Such trajectories pose a risk of misleading the policy by ignoring the problematic outcomes of executing visual operations.

To mitigate this issue and ensure complete control over the synthesized trajectories, we employ a template-based synthesis approach. As shown in Fig. 3 (b), this template structures a pixel-space reasoning trajectory as a sequence: initial analysis of the entire visual input, followed by triggering specific visual operations to extract fine-grained details, subsequent analysis of these detailed visual cues, and ultimately arriving at the final answer. To synthesize a trajectory according to this template, we utilize the reference visual cue associated with each vision-language query. We first prompt GPT-4o to generate a textual description summarizing the entire visual input. Then, leveraging the reference visual cue, we prompt GPT-4o for a more detailed textual analysis focusing specifically on that cue. By composing these textual thinking segments and incorporating the visual operation targeting the reference visual cue, we obtain a pixel-space reasoning trajectory that effectively interleaves textual reasoning with required visual operations.

In addition to these basic **single-pass trajectories** that help the policy understand the effective utilization of visual operations, we also synthesize **error-induced self-correction trajectories**. These are designed to preserve and foster the policy's ability to properly react to unexpected inputs or errors during execution. As illustrated in Fig. 4, we synthesize such trajectories by deliberately choosing incorrect or improper visual cues, such as an irrelevant video frame or overly large image regions, for reaching the correct answer. We then insert the visual operations and textual thinking segments for these distracting visual cues before introducing the correct reference visual cues, thus simulating self-correction behaviors in error-induced trajectories.

**Warm-Start Instruction Tuning.** We include two primary types of pixel-space reasoning trajectories in our training data: single-pass and error-induced self-correction trajectories. We also include textual reasoning trajectories for vision-language queries that do not necessitate fine-grained visual analysis. This mixed data composition allows the policy to adaptively employ pixel-space reasoning only when necessary. We employ the standard Supervised Fine-Tuning (SFT) loss for training. However,

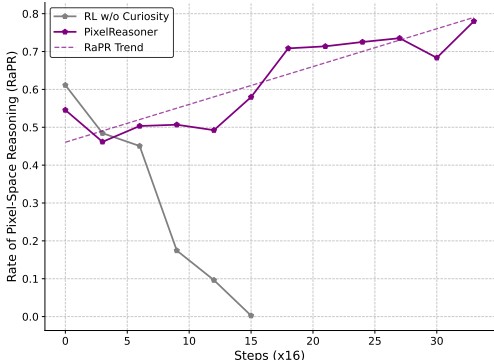

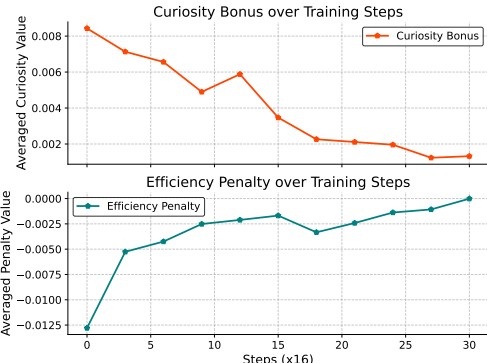

Figure 5: RL Requires Incentives to Explore Pixel-space Reasoning. Without proper incentives, the policy learns to bypass the nascent pixel-space reasoning, resulting in declining RaPR.

Figure 6: The Training Trend of our Curiosity-Driven Reward Scheme. We leverage curiosity bonus to encourages exploration and efficiency penalty to punish excessive visual operations.

we apply loss masks to tokens that represent either execution outputs from visual operations or the specifically designated erroneous visual operations within the self-correction trajectories. Masking the erroneous operations prevents the policy from learning to execute the incorrect actions.

## 4 Curiosity-Driven Reinforcement Learning

The warm-started model typically suffers from a disparity in its capabilities: proficient textual reasoning versus nascent pixel-space reasoning. This inherent imbalance creates a "learning trap" that impedes the development of pixel-space reasoning, stemming from two synergistic issues. Firstly, the model's initial limited mastery over visual operations frequently leads to failure or incorrect outputs, resulting in a higher incidence of negative feedback compared to text-mediated reasoning. Secondly, a significant portion of training queries does not rigorously demand visual processing for a correct response, allowing the model to ignore the outcomes of visual operations or default to its stronger textual reasoning. This interplay fosters a detrimental cycle where initial failures discourage further attempts, leading to the premature abandonment of exploring and mastering visual operations. As shown in Fig. 5, when training the Warm-Start Model with standard RL [DeepSeek-AI et al., 2025, Wang et al., 2025] without proper incentives, the policy learns to bypass the nascent visual operations.

To break this cycle, we propose a curiosity-driven reward scheme to incentivize sustained exploration of pixel-space reasoning, inspired by curiosity-driven exploration in conventional RL [Pathak et al., 2017]. Instead of relying solely on extrinsic rewards for correctness, this curiosity bonus specifically incentivizes the act of attempting pixel-space operations. By intrinsically rewarding such active practice, we aim to bolster the model's nascent visual skills and counteract the discouragement of exploration that arises from early operational failures and the associated negative feedback. This mirrors how a child, driven by curiosity, might repeatedly attempt a difficult motor task, learning from each attempt, rather than immediately defaulting to an easier, already mastered skill.

Specifically, we formalize this objective as a constrained optimization problem. Let $\mathbf{1}_{\mathrm{PR}}(\mathbf{y})$ denotes the indicator function of response $\mathbf{y}$ utilizing pixel-space reasoning, and $\mathbf{n}_{\mathrm{vo}}(\mathbf{y})$ represent the number of visual operations. The goal is to maximize the expected correctness outcome, subject to two critical constraints meticulously designed to cultivate the pixel-space reasoning:

$$\max_{\theta} \mathbb{E}\left[r(\mathbf{x}, \mathbf{y}) \Big| \mathbf{x} \sim \mathcal{D}, \mathbf{y} \sim \pi_\theta(\mathbf{y}|\mathbf{x})\right] \tag{1}$$

$$\text{subject to} \quad \mathrm{RaPR}(\mathbf{x}) \doteq \mathbb{E}\left[\mathbf{1}_{\mathrm{PR}}(\mathbf{y}) \Big| \mathbf{y} \sim \pi_\theta(\mathbf{y}|\mathbf{x})\right] \geq \mathrm{H}, \mathbf{n}_{\mathrm{vo}}(\mathbf{y}) \leq \mathrm{N} \tag{2}$$

Here, the first constraint concerns the *Rate of Pixel-space Reasoning (RaPR)* (pronounced "rapper") triggered for a query $\mathbf{x}$. We mandate that this rate, averaged over rollouts $\mathbf{y}$ of query $\mathbf{x}$, must be no less than a predefined threshold H. This encourages the policy to consistently attempt pixel-space reasoning across a significant proportion of queries, acting as a directive to explore this less familiar reasoning path. The second constraint imposes an upper bound N on the number of visual operations

used in any individual response. This ensures that while exploration is encouraged, it remains computationally efficient and does not lead to overly complex or protracted visual processing for individual responses.

This constrained optimization problem can be transformed into an unconstrained problem via Lagrangian Relaxation [Lemaréchal, 2001], resulting in a single reward function. This technique is commonly employed in constrained RL [Achiam et al., 2017, Wang et al., 2022, 2023]. The transformation yields the following modified reward function $r'(\mathbf{x}, \mathbf{y})$, detailed in the appendix:

$$r'(\mathbf{x}, \mathbf{y}) = r(\mathbf{x}, \mathbf{y}) + \alpha \cdot r_{\text{curiosity}}(\mathbf{x}, \mathbf{y}) + \beta \cdot r_{\text{penalty}}(\mathbf{y}), \tag{3}$$

$$\text{where} \quad r_{\text{curiosity}}(\mathbf{x}, \mathbf{y}) = \max(\text{H} - \text{RaPR}(\mathbf{x}), 0) \cdot \mathbf{1}_{\text{PR}}(\mathbf{y}) \tag{4}$$

$$r_{\text{penalty}}(\mathbf{y}) = \min(\text{N} - \mathbf{n}_{\text{vo}}(\mathbf{y}), 0) \tag{5}$$

The modified reward incorporates two additional terms. The first term $r_{\text{curiosity}}(\mathbf{x}, \mathbf{y})$ serves as the core of our curiosity mechanism. It provides an intrinsic reward that directly encourages the model to satisfy its "curiosity" about pixel-space operations, especially for queries where it has a low history of attempting them. Akin to infants curious about and exploring unseen environments or novel interactions, this term credits response $\mathbf{y}$ a bonus for employing pixel-space reasoning when the adoption of pixel-space reasoning, $\text{RaPR}(\mathbf{x})$, is below a target threshold H. This curiosity bonus effectively lowers the activation energy for trying the visual operations, making the model more "inquisitive" and willing to venture into less certain reasoning paths. The second term, $r_{\text{penalty}}(\mathbf{y})$, acts as an efficiency penalty at the response level, penalizing redundancy in visual operations by considering the number of visual operations performed, $\mathbf{n}_{\text{vo}}(\mathbf{y})$, relative to a desired maximum N.

The coefficients $\alpha \geq 0$ and $\beta \geq 0$ are non-negative Lagrangian multipliers. These multipliers can be tuned automatically, for instance, via dual gradient descent [Bishop and Nasrabadi, 2006, Wang et al., 2020], or set as pre-defined hyperparameters [Wang et al., 2022]. In our experiments, we adopt the latter approach for simplicity. We provide a concrete example in the appendix to illustrate how these hyperparameters reflect our desired properties of the policy.

This reward scheme offers a dynamic reward mechanism that automatically tune the exploration bonus as training proceeds. As illustrated in Fig. 6, the curiosity bonus will naturally diminishes as the policy explores more on pixel-space reasoning. This prevents the policy policy from reward hacking – overly relying on the exploration bonus regardless of final correctness.

## 5  Experiments

In this section, we first outline the training and evaluation settings. We then examine the effectiveness of pixel-space reasoning, and study the key factors for cultivating pixel-space reasoning.

**Training Data and Evaluation Settings.** Utilizing the data curation pipeline outlined in Section 3, we assembled a dataset of 7,500 trajectories for warm-start instruction tuning. This dataset includes 5,500 pixel-space reasoning trajectories synthesized using GPT-4o, spanning domains such as images, webpages, and videos. We also include 2,000 text-space reasoning trajectories to balance the use of visual operations. During RL, we construct 15,000 queries from our SFT dataset, InfographicVQA [Mathew et al., 2021], and publicly available datasets [Xu et al., 2025, Wu et al., 2024]. Refer to the appendix for a comprehensive view of the dataset compositions.

We evaluated our model and other baselines on four representative multimodal benchmarks using greedy decoding: TallyQA, V*, InfographicVQA, and MVBench. This selection offers a wide spectrum of visual understanding tasks, from fine-grained object recognition to high-level reasoning in both static and dynamic scenarios. Specifically, V* (V-Star) [Wu and Xie, 2024] evaluates multimodal large language models (MLLMs) on their ability to process high-resolution, visually complex images and focus on fine-grained visual details. TallyQA [Acharya et al., 2019] consists of questions that require reasoning over object quantities, often demanding the model to locate, differentiate, and tally objects across complex scenes. MVBench [Li et al., 2024a] is a comprehensive benchmark designed to evaluate multimodal large language models (MLLMs) on their temporal understanding capabilities across 20 challenging video tasks, necessitating reasoning beyond static image analysis. InfographicVQA [Mathew et al., 2021] evaluates the model's ability to understand complex infographic images that blend textual and visual content, including charts, diagrams, and annotated images. Success on this benchmark requires parsing layout, reading embedded text, and linking visual elements with semantic meaning.

Table 1: Our main results on the four evaluated benchmarks.

| Model | Size | V* Bench | TallyQA-complex | MVBench-test | InfoVQA-test |
| Metric | | Acc | Acc | Acc | ANLS |
|---|---|---|---|---|---|
| Models w/o Tools | | | | | |
| GPT-4o | - | 62.8 | 73.0 | **64.6** | 80.7 |
| Gemini-2.0-Flash | - | 73.2 | 73.8 | - | **86.5** |
| Gemini-2.5-Pro | - | 79.2 | **74.0** | - | 84.0 |
| Qwen2.5-VL | 7B | 70.4 | 68.6 | 63.8 | 80.7 |
| Video-R1 | 7B | 51.2 | 42.6 | 63.9 | 67.9 |
| LongLLava | 13B | 68.5 | 64.6 | 54.6 | 65.4 |
| Gemma3 | 27B | 62.3 | 54.3 | 56.8 | 59.4 |
| Models with Tools | | | | | |
| Visual Sketchpad (GPT-4o) | - | 80.4 | - | - | - |
| IVM-Enhance (GPT-4V) | - | **81.2** | - | - | - |
| PaLI-3-VPD | 5B | 70.9 | - | - | - |
| SEAL | 7B | 74.8 | - | - | - |
| PaLI-X-VPD | 55B | 76.6 | - | - | - |
| Ours (Initialized from Qwen2.5-VL-7B) | | | | | |
| Pixel-Reasoner | 7B | **84.3** | 73.8 | 67.8 | 84.0 |
| Ablation Baselines (Ablated from Pixel-Reasoner) | | | | | |
| Warm-Start Model (w/o RL) | 7B | 79.0 | 67.9 | 59.0 | 74.3 |
| RL w/o Curiosity | 7B | 81.1 | 71.8 | 66.4 | 80.7 |
| RL w/o Warm-Start | 7B | 81.7 | 72.2 | 65.6 | 81.2 |
| RL w/o Correction-Data | 7B | 80.1 | 69.8 | 63.6 | 78.2 |

**Compared Models and Implementation.** We compare against a wide range of models.

- Models without Tools: We include GPT-4o [Hurst et al., 2024] and Gemini-2.0-Flash [Team et al., 2024b] and Gemini-2.5-Pro [Team et al., 2024a]. These models do not have access to tools and simply answer with chain-of-thought. We include Qwen2.5-VL [Bai et al., 2025], Gemma3 [Team et al., 2025] to show the general VLMs' performance. We also compare with RL-based VLM model Video-R1 [Feng et al., 2025] due to the similar algorithm. We further include LongLlava [Wang et al., 2024] because it aims to scale up image input to deal with high-resolution images (V*) and long video sequence (MVBench).
- Models with Tools: We include Visual Sketchpad [Hu et al., 2024a], which empowers the GPT-4o to use different tools like zoom-in, depth, etc. We also include Instruction-Guided Visual Masking [Zheng et al., 2024], which highlights desired region in a given image. Finally, we add Visual-Program-Distillation (VPD) [Hu et al., 2024b], which aims to distill tools reasoning into closed-source VLMs like PaLI. These models are specialized in V* Bench. We include SEAL [Wu and Xie, 2024] from original V* Bench paper, which utilizes visual guided search tool to augment high-resolution image understanding.

Pixel-Reasoner was trained on $8 \times A800(80G)$ GPUs, using Open-R1 and OpenRLHF for instruction tuning and reinforcement learning respectively. We adopt GRPO [DeepSeek-AI et al., 2025] with selective sample relay due to vanishing advantages [Wang et al., 2025]. We include training details in the appendix, and will release code, models, and data to support reproducibility.

## 5.1 Main Results

Table 1 shows that Pixel-Reasoner achieves the highest open-source results across all four benchmarks. Remarkably, Pixel-Reasoner, at a mere 7B parameters, not only surpasses substantially larger open-source models like the 27B Gemma3 across all benchmarks, but also outperforms specialized models that depend on external tools, such as IVM-Enhance (GPT-4V). Furthermore, Pixel-Reasoner's exceptional capabilities extend to outperforming leading proprietary models, evidenced by its significant 5.1 percentage point lead over Gemini-2.5-Pro on V-star Bench (84.3 vs 79.2), and achieving the overall highest scores amongst all models listed. We observe that RL training

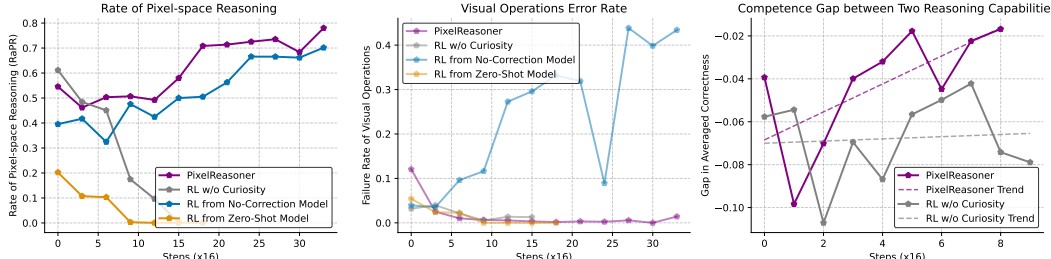

Figure 7: Trainig Dynamics of Ablation Baselines. During RL training, different baselines show different trends in triggering pixel-space reasoning (Left), and the error rate of utilizing visual operations (Middle). Our curiosity-driven reward scheme effectively cultivates pixel-space reasoning by actively practicing and enhancing this nascent ability, as evidenced by the narrowed gap in return between the two reasoning paradigms (Right).

is monumental to the great results. Our ablation model "Warm-Start Model (w/o RL)" has a lower performance than the original checkpoint Qwen2.5-VL on many benchmarks. After RL training, the performance skyrockets to SoTA level. This reflects the necessity of RL training to cultivate pixel-level reasoning.

The performance profoundly underscores the significant potential of our proposed pixel-space reasoning paradigm. This potential is further highlighted when comparing `Pixel-Reasoner` with the ablation baselines, "RL w/o Curiosity" and "RL w/o Warm-Start". Due to insufficient incentives and limited proficiency in utilizing visual operations, these baselines ultimately default to text-space reasoning, resulting in a significant performance drop of 2.5 points on average across benchmarks. These empirical gains verify the effectiveness of pixel-space reasoning – by enabling the model to proactively engage with visual operations, this new reasoning paradigm facilitates a more precise visual understanding and consequently, a stronger reasoning capability.

## 5.2 Key Factors for Cultivating Pixel-Space Reasoning

This section investigates two critical factors in fostering pixel-space reasoning through RL: first, the policy's proficiency in utilizing visual operations lays the foundation for RL, and second, the role of incentives in encouraging the adoption of pixel-space reasoning during RL. To gain deeper insights into the results in Tab. 1, we analyze the training dynamics of various baselines, shown in Fig. 7. Specifically, the left panel shows the proportion of training rollouts that employ pixel-space reasoning strategies, while the middle panel indicates the error rate associated with the execution of visual operations. The right panel compares the expected correctness between the two reasoning paradigms, highlighting the disparity in the two capabilities over training time. In addition, the appendix provides concrete examples of trajectories that bypass pixel-space reasoning, illustrating "the learning trap."

**Effective Utilization of Visual Operations Requires Instruction Tuning.** A crucial finding is that meticulous warm-start instruction tuning is essential for enhancing the policy's mastery of visual operations and its capacity for self-correction. To demonstrate this, we analyze the RL training dynamics originating from three distinct instruction-tuned models: (a) The Warm-Start Model that undergoes the proposed warm-start instruction tuning phase. (b) The No-Correction Model is tuned using single-pass expert trajectories but without the error-induced self-corrective trajectories. (c) The Zero-Shot Model is Qwen2.5-VL-Instruct with zero-shot prompts of available visual operations.

The training dynamics reveal distinct outcomes:

- **The Zero-Shot Model** (orange lines) commences with a low RaPR of approximately 20%, which progressively declines during RL and reaches zero. This initial low propensity to trigger visual operations provides insufficient practice on visual operations. Consequently, the model receives lower expected returns from its nascent pixel-space reasoning compared to its more established textual reasoning, leading to a diminishing RaPR. This illustrates how limited initial proficiency in visual operations can create a detrimental cycle, hindering the development of pixel-space reasoning. Its error rate for visual operations also remains low due to their minimal usage.
- **The No-Correction Model** (blue lines), trained solely on single-pass expert trajectories, initially exhibits an increase in RaPR, suggesting a propensity for attempting visual operations. However, this trend is quickly overshadowed by a significant and persistent rise in the failure rate of

these operations. This elevated error rate points to the model's inability to effectively respond to unexpected or erroneous outcomes from visual tasks. This deficiency stems directly from the absence of error-induced self-correction trajectories during its instruction-tuning phase. Consequently, the policy increasingly relies on pixel-space reasoning while simultaneously ignoring the outcomes of visual operations and favoring textual reasoning. Interestingly, we observe that the resulting reasoning trajectories involve error messages from visual operations but can still arrive at a correct answer. This indicates reward hacking: the policy earns curiosity bonus by superficially executing visual operations, and meanwhile it also earns correctness reward by essentially relying on textual reasoning to arrive at the final answer.

- **The Warm-Start Model** (purple lines as "PixelReasoner"), in constrast, serves as the foundation for RL and is appropriately incentivized, it enables the successful cultivation of pixel-space reasoning without exhibiting excessive error rates in visual operations. This underscores the importance of the comprehensive instruction tuning provided by the warm-start phase.

**Cultivation of Novel Reasoning Capabilities Requires Incentives.** To evaluate the impact of incentives, we compare the RL training dynamics starting from the Warm-Start Model, both with and without curiosity-driven incentives.

- **Standard RL without curiosity** (grey lines). The curve shows a consistent decrease in the utilization of visual operations (RaPR), from around 0.55 to 0 in 240 gradient steps. This decline occurs because, without a specific impetus to explore, the policy favors its more developed textual reasoning over the initially less competent pixel-space reasoning. The failure rate of visual operations remains low as their usage diminishes.
- **Our Model** (purple lines), which also starts from the Warm-Start Model but incorporates a curiosity-driven exploration bonus, demonstrates a more complex and ultimately successful trajectory. Initially, `Pixel-Reasoner` exhibits a decrease in RaPR in the first 50 gradient steps, and then plateaus for around 150 steps. During this stage, the policy is compelled by the curiosity bonus to continue exploring pixel-space reasoning, despite its relative inferiority compared to textual reasoning (as shown in Fig. 7 (Right)). Not until 200 gradient steps, the policy starts to effectively leverage the benefits of pixel-space reasoning. Its RaPR proactively and substantially increases, accompanied by a low and stable failure rate for visual operations. This indicates that the combination of the robust Warm-Start instruction tuning and the curiosity-driven incentive allows the policy to not only explore but also master the new pixel-space reasoning capability. Also note that `Pixel-Reasoner` exhibit relatively high RaPR of 80% due to the high proportion of visually intensive tasks in training queries. We provide curves of test sets in the appendix.

## 6  Related Work

**Post-Training for Vision-Language Models.** Post-training techniques, such as instruction tuning and reinforcement learning, are critical for adapting large Vision-Language Models (VLMs) to complex tasks beyond initial pre-training. LLaVA [Liu et al., 2023], Llava-OV [Li et al., 2024b], Infinity-MM [Gu et al., 2024], and MAmmoTH-VL [Guo et al., 2024] has shown that scaling instruction tuning datasets and increasing task diversity significantly enhances VLM generalization across various multimodal benchmarks.

Recently, a growing body of work applies RL to the multimodal domain [Deng et al., 2025, Huang et al., 2025, Feng et al., 2025]. These approaches typically employ multi-stage pipelines, starting with SFT on costly data distillation and then applying RL to further refine the model's reasoning capabilities. VL-Rethinker [Wang et al., 2025] investigates more direct RL approaches to foster slow-thinking in VLMs, and introduced selective sample replay (SSR) to counteract the vanishing advantages problem in GRPO.

**Vision-Language Models with Tools.** Recent research has explored augmenting VLMs with external tools or enabling them to perform pixel-level operations on inputs. Chain-of-Manipulation [Qi et al., 2025], Visual-Program-Distillation (VPD) [Hu et al., 2024b] focus on training models to effectively utilize tools or distill tool-based reasoning. Visual Sketchpad [Hu et al., 2024a] equips models, such as GPT-4o, with tools like depth perception and Python plotting. Models like o3 [Jaech et al., 2024b] demonstrate an ability to "think with images" by dynamically applying operations like zooming or flipping to improve visual understanding. Specific tools such as Instruction-Guided Visual Masking [Zheng et al., 2024] and visual guided search [Wu and Xie, 2024] has been integrated into these frameworks.

## 7 Conclusion

In this paper, we show how to incentivize the pixel-space reasoning from an existing vision-language models for the first time. Our warm-start instruction tuning and curiosity-driven RL are both essential to achieve the state-of-the-art performance. However, our work is currently still limited to two primary operations, which is insufficient for broader tasks. Our framework is easily extensible to other operations like depth map, image search, etc. In the future, the community can work together to enrich the visual operations to enhance the pixel-space reasoning in VLMs.

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

# Appendix

## A  Limitations

In this work, we pose a fundamental rethinking of vision-language reasoning, and introduce the concept of pixel-space reasoning. While we show the effectiveness of our approach to cultivating pixel-space reasoning, this improvement is still bottleneck-ed by limited data that spans across tasks and contents. In addition, we focus on two specific visual operations to handle primary media formats of images and videos. In the future, we endeavor to include more visual operations and examine the effectiveness of pixel-space reasoning on more diverse collections of tasks.

## B  Derivations of Curiosity-Driven Reward

The primary objective is to maximize the expected correctness outcome, formalized as a constrained optimization problem. Let $r(\mathbf{x}, \mathbf{y})$ be the original correctness reward for a query $\mathbf{x}$ and response $\mathbf{y}$. The policy generating responses is denoted by $\pi_\theta(\mathbf{y}|\mathbf{x})$.

The optimization problem is:

$$\max_\theta \quad \mathbb{E}\left[r(\mathbf{x}, \mathbf{y})\Big| \mathbf{x} \sim \mathcal{D}, \mathbf{y} \sim \pi_\theta(\mathbf{y}|\mathbf{x})\right] \tag{6}$$

$$\text{subject to} \quad C_1(\theta; \mathbf{x}) \equiv \text{RaPR}(\mathbf{x}) - \text{H} \geq 0 \tag{7}$$

$$C_2(\mathbf{y}) \equiv \text{N} - \mathbf{n}_{\text{vo}}(\mathbf{y}) \geq 0 \tag{8}$$

where:

- $\text{RaPR}(\mathbf{x}) \doteq \mathbb{E}\left[\mathbf{1}_{\text{PR}}(\mathbf{y})\Big|\mathbf{y} \sim \pi_\theta(\mathbf{y}|\mathbf{x})\right]$ is the Rate of Pixel-space Reasoning for query $\mathbf{x}$.

- H is a predefined minimum threshold for $\text{RaPR}(\mathbf{x})$.

- $\mathbf{n}_{\text{vo}}(\mathbf{y})$ is the number of visual operations in response $\mathbf{y}$.

- N is a predefined upper bound on $\mathbf{n}_{\text{vo}}(\mathbf{y})$.

Constraint (7) is an expectation-level constraint for a given query $\mathbf{x}$, while constraint (8) applies to each individual response $\mathbf{y}$.

To incorporate these constraints into the objective, a common technique is the method of Lagrangian Relaxation. For a maximization problem, this typically involves subtracting terms proportional to the constraint violations (when constraints are written as $g(x) \leq 0$) from the original objective function $r(\mathbf{x}, \mathbf{y})$. If we rewrite our constraints as $g_1(\theta; \mathbf{x}) \equiv \text{H} - \text{RaPR}(\mathbf{x}) \leq 0$ and $g_2(\mathbf{y}) \equiv \mathbf{n}_{\text{vo}}(\mathbf{y}) - \text{N} \leq 0$, the standard Lagrangian modification to the per-instance reward would be:

$$r_{\text{Lagrangian}}(\mathbf{x}, \mathbf{y}; \theta) = r(\mathbf{x}, \mathbf{y}) - \lambda_1(\text{H} - \text{RaPR}(\mathbf{x})) - \lambda_2(\mathbf{n}_{\text{vo}}(\mathbf{y}) - \text{N}) \tag{9}$$

where $\lambda_1, \lambda_2 \geq 0$ are Lagrange multipliers. The overall optimization objective would then be to maximize $\mathbb{E}\left[r_{\text{Lagrangian}}(\mathbf{x}, \mathbf{y}; \theta)\right]$ with respect to $\theta$, and to minimize with respect to the multipliers.

However, directly applying this standard formulation has two problems. Firstly, this formulation has an over-satisfaction issue. The term $-\lambda_2(\mathbf{n}_{\text{vo}}(\mathbf{y}) - \text{N})$ would provide a positive reward if $\mathbf{n}_{\text{vo}}(\mathbf{y}) < \text{N}$ (i.e., the constraint is "over-satisfied"), potentially encouraging the policy to use far fewer visual operations than necessary. Secondly, the term $-\lambda_1(\text{H} - \text{RaPR}(\mathbf{x}))$ operates on the expectation-level and does not properly reward individual responses $y \sim \pi_\theta$.

Therefore, we adopt the following modified reward function:

$$r'(\mathbf{x}, \mathbf{y}) = r(\mathbf{x}, \mathbf{y}) + \alpha \cdot \max(\text{H} - \text{RaPR}(\mathbf{x}), 0) \cdot \mathbf{1}_{\text{PR}}(\mathbf{y}) + \beta \cdot \min(\text{N} - \mathbf{n}_{\text{vo}}(\mathbf{y}), 0) \tag{10}$$

where $\alpha \geq 0, \beta \geq 0$ are fixed hyperparameters.

This formulation offers several benefits. Firstly, the clipping mechanism addresses the over-satisfaction issue while preserving equivalence to the original constrained objective [Wang et al., 2022]. The clipping ensures the penalties are active only when the respective constraints are violated, otherwise the penalties are zero, thus avoiding over-statisfaction.

Secondly, this structure allows $\alpha, \beta$ to be treated as fixed hyperparameters. In standard Lagrangian methods (Eq. 9), multipliers are often dynamically adjusted; for example, Karush-Kuhn-Tucker (KKT) conditions imply that multipliers for inactive constraints (those satisfied with slack) are zero. The clipping zeros out the penalties when constraints are satisfied, thereby obviating the need for dynamic adjustment of $\alpha, \beta$ based on constraint satisfaction levels.

In addition, the inclusion of the indicator $\mathbf{1}_{\mathrm{PR}}(\mathbf{y})$ converts the query-level expectation constraint into a response-level reward. Intuitively, this term acts as a targeted incentive: it rewards the specific behavior of engaging in pixel-space reasoning precisely when the average rate of such reasoning is below the desired threshold. The multiplier $\alpha \geq 0$ scales this incentive. It provides an implicit penalty for missing out on the potential bonuses the policy could have earned by employing pixel-space reasoning.

## C  Data and Training Details

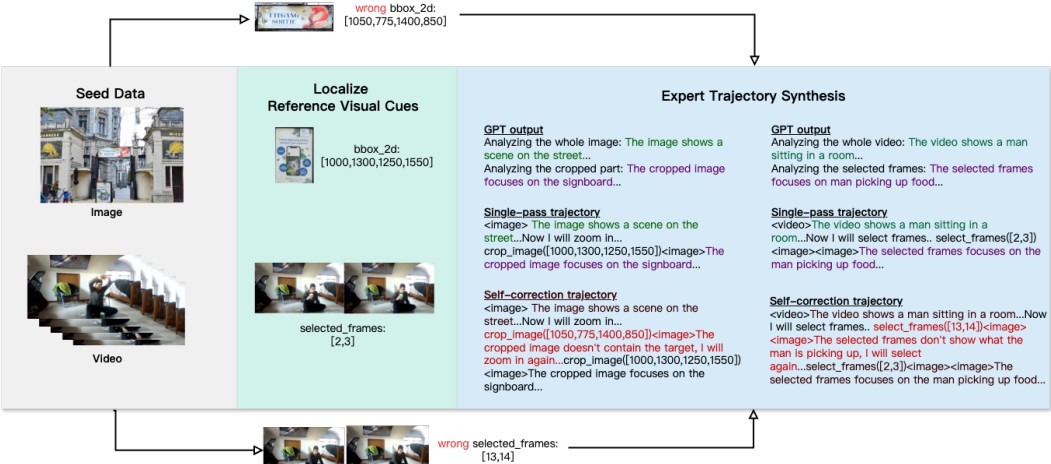

Figure 8: A detailed illustration of our data generation pipeline.

### C.1  Protocols of Visual Operations

We include two primary visual operations: cropping an image and selecting frames from a video.

**CropImage**   This operation allows the model to zoom in on a specific region of an image by providing a bounding box. The input includes a two-dimensional bounding box bbox_2d—a list of numeric coordinates $[x_1, y_1, x_2, y_2]$ constrained within the image dimensions—and a target_image index indicating which image to operate on (indexed from 1, where 1 refers to the original image). This operation helps the model focus on fine-grained details.

**SelectFrames**   This operation enables the model to select a subset of frames from a video. The input target_frames is a list of integer indices specifying which frames to extract from a 16-frame sequence, with a limit of no more than 8 frames. This allows the model to focus on key temporal moments relevant to the query.

### C.2  Instruction Tuning Data

**Details of Seed Datasets** We selected datasets based on two key attributes: high visual complexity requiring fine-grained analysis, and the presence of explicit annotations that can serve as targets or anchors for visual operations. Based on these criteria, our data sources include:

- **SA1B** [Kirillov et al., 2023]: A large-scale dataset of high-resolution natural scenes offering rich visual detail and complexity.

- **FineWeb** [Ma et al., 2024]: Consists of webpage screenshots paired with Question-Answering (QA) instances and precise bounding box annotations for answer regions, offering explicit spatial targets for visual analysis.
- **STARQA** [Wu et al., 2024]: Provides video data with QA pairs and annotated temporal windows indicating relevant visual contents for answers, offering both visual and temporal context for potential video-specific operations.

**Detailed Data Pipeline Illustration.** As the Fig. 8 depicts, after we obtain reference visual cues from seed data, we input both the whole HR image or video and the corresponding localized reference visual cues to gpt. Then we use template-based method to extract whole visual input analysis and local detailed analysis before we concatenate the whole analysis, localized reference visual cue and the partial analysis to form the single-pass trajectory. We utilize the reference visual cue to get the wrong visual cues to insert in the obtained single-pass trajectories to get self-correction trajectory.

**Single-pass and Self-correction Data Synthesis Details**

| Category | Trajectory Type | Proportion |
|---|---|---|
| Image | single-pass | 30% |
| | Recrop once | 20% |
| | Recrop twice | 20% |
| | Further zoom-in | 30% |
| Video | single-pass | 90% |
| | Reselect | 10% |

Table 2: Self-correction trajectory types and corresponding proportions.

Here `single-pass` means no error is inserted in the trajectory. `Recrop once` means we randomly select a bbox that has no intersection with the reference visual cue and insert it before the correct visual operation. `Recrop twice` means we randomly select 2 bboxes that have no intersection with the reference visual cue and insert them sequentially before the correct visual operation. `Further zoom-in` means we select an inaccurate bbox that contains the reference visual cue but is excessively larger than it, and we insert it before the correct visual operation. `Reselect` means we sample frame indexes that have no intersection with the reference visual cue's frame indexes, and we insert it before the correct visual operation.

### C.3 Training Details

**Implementation Details**. For Instruction Tuning, we adapt the Open-R1 code to implement SFT loss with loss masks. For RL, we implement based on OpenRLHF. We adopt GRPO [DeepSeek-AI et al., 2025] with selective sample replay [Wang et al., 2025], because we witness significant issues of vanishing advantages. As shown in Fig. 9, our reward scheme incorporates curiosity bonus and efficiency penalty in addition to correctness rewards, which provides more variance in rewards. However, the ratio of queries that suffer from reward uniformity steadily increases to 90% as training progresses, leading to a drastic plunge in performance evidenced by the ratios of "response-all-incorrect" queries. During RL training, we employed a near on-policy RL paradigm, where the behavior policy was synchronized with the improvement policy after every 512 queries, which we define as an episode. The replay buffer for SSR persisted for the duration of each episode before being cleared. For each query, we sampled 8 responses. The training batch size was set to 256 query-response pairs. Our 7B model is trained on $4 \times 8$ sets of A800 (80G) for 20 hours .

**Training Hyperparameters.** For Instruction Tuning, we use a batch size of 128. The learning rate is $1e^{-6}$ with 10% warm up steps. For RL, we set employ a cosine learning rate schedule with initial learning rate $1e^{-6}$ and 3% warm up iterations. During RL training, we sample 8 trajectories per training query and set hyperparameters to $\alpha = 0.5$, $\beta = 0.05$, H $= 0.3$, and N $= 1$. This configuration reflects our objectives: the threshold H $= 0.3$ encourages the policy to utilize pixel-space reasoning in approximately 30% of responses generated for a given query, while N $= 1$ promotes efficiency by favoring responses that require at most one visual operation. Under these parameters, a response can receive a maximum exploration bonus of approximately

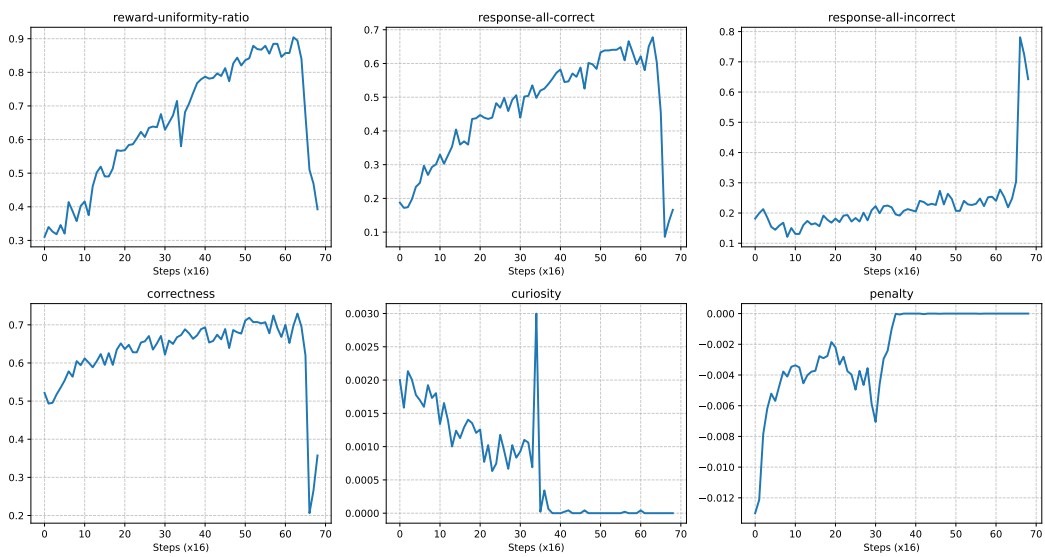

Figure 9: Trainig Dynamics of RL without SSR. The ratio of reward uniformity steadily saturates to 90%.

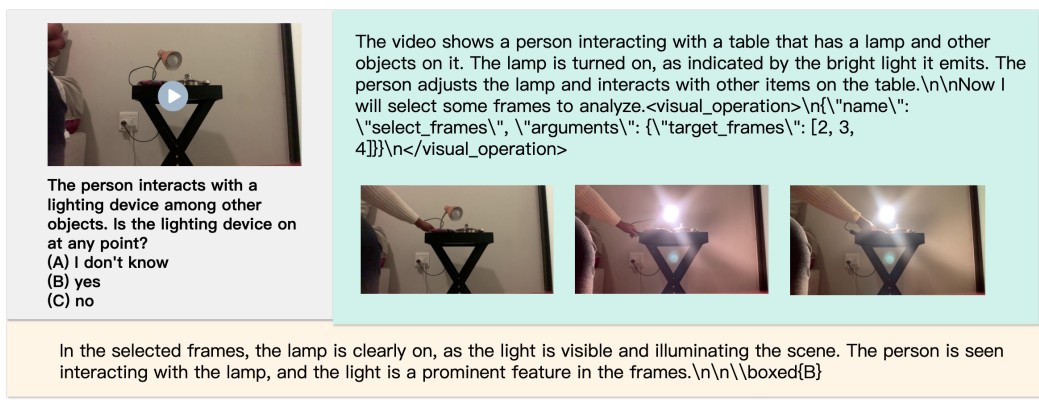

Figure 10: `Pixel-Reasoner` effectively captures key frames that are most informative for solving the query.

$0.5 \times (0.3 - 1/8) \approx 0.0875$, while each additional visual operation beyond the first incurs a penalty of $-0.05$.

# D    Additional Analysis

## D.1    Statistics

**Qwen2.5-VL-Instruct shows limited Zero-Shot Ability in utilizing novel visual operations.** We include the protocols of visual operations in the system prompt for Qwen2.5-VL-Instruct. Initially, it invokes visual operations in 20.2% training rollouts, where 40.6% of them incurs error and 36.2% leads to incorrect answer. This results in 23.2% average accuracy when utilizing pixel-space reasoning, in contrast to 49.5% average accuracy when utilizing textual reasoning.

**RaPR of `Pixel-Reasoner` on the Evaluation Benchmarks.** Our `Pixel-Reasoner` adaptively triggers pixel-space reasoning with a portion of 78.53% on V-Star, 57.78% on TallyQA-Complex, 58.95% on InfographicsVQA, and 66.95% on MVBench.

## D.2    Case Analysis

We show successful trajectories in Fig. 11 and Fig. 10. We also provide concrete examples to show the failure modes when the policy suffers from "the learning trap".

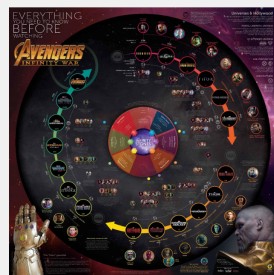

In which movie does Tony Stark create an armour to save his life?

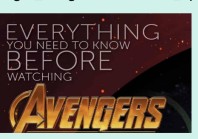

The image is an infographic titled \"Everything You Need to Know Before Watching Avengers: Infinity War.\" It provides a timeline of the Marvel Cinematic Universe, showing the chronological order of movies, characters, and events. The infographic also highlights the Infinity Stones and their significance. Now I will zoom in to look clearer at the section about \"Tony Stark creating an armor to save his life. <visual_operation>{\"name\": \"crop_image\", "arguments": {\"bbox_2d\": [225, 100, 675, 300], \"target_image\":1}}</visual_operation>

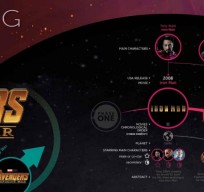

The cropped image focuses on the title and the beginning of the infographic. The section about \"Tony Stark creating an armor to save his life\" is not visible in this cropped image. I need to zoom in further to get a clearer view of the relevant part.<visual_operation>\n{\"name\": \"crop_image\", \"arguments\": {\"bbox_2d\": [1012, 300, 1387, 600], \"target_image\":1}} </visual_operation>

The cropped image focuses on the \"Iron Man\" section. The text states: \"Tony Stark creates an armor to save his life, then evolving it into Iron Man.\" The year 2008 is also mentioned, which corresponds to the movie \"Iron Man.\"\n\n\\boxed{Iron Man}

Figure 11: `Pixel-Reasoner` effectively identifies the relevant region within the infographic and performs targeted analysis, either by refining the crop or directly answering the query.

**Failure mode 1: Hallucination.** In the example below, the policy hallucinates as if the operations are successfully executed.

```
The video shows a table with various items, including vegetables and a bowl.
The person is seen interacting with the vegetables, specifically placing them into the
bowl. At one point, the person is seen shuffling some letters on the table. The video
does not clearly show the sequence or order of the letters throughout the video.
Now I will select some frames to look clearer at the sequence of the letters.

Execution error:max() arg is an empty sequence

The cropped video frames show the person continuing to prepare the salad and shuffle the
letters. The letters appear to be in a different arrangement compared to the beginning
of the video.\n\n\\boxed{B}
```

**Failure mode 2: No-Reaction.** In the example below, the policy ignores the error message and proceeds with textual reasoning.

```
The video shows a person in a kitchen. The person interacts with objects on the stove,
    including a pot. At one point, the person appears to handle an object on the
    counter and then moves away from the stove area. The sequence of actions suggests
    the person might have placed an item down or completed an action involving the
    counter.\n\nNow I will select some frames to look clearer at which object was put
    down by the person.

Execution error:max() arg is an empty sequence

It seems there was an issue with selecting frames. Let me describe the scene instead:\n\
    nThe person in the video is standing near the stove with a pot on it. After
    interacting with the pot, the person moves to the counter area. It appears the
    person may have placed an object (like a box or a paper/notebook) down on the
    counter before moving away. The object is not visible in the provided frames, but
    the sequence of actions suggests it was placed there.\n\n\\boxed{C}
```

# E   Prompts

## E.1   Prompts Used in RL Training

During RL training, we append the following prompt after the user query.

```
{question}
\n\nGuidelines: Understand the given visual information and the user query. Determine if
```

```
it is beneficial to employ the given visual operations (tools). For a video, we can look
closer by `select_frames`. For an image, we can look closer by `crop_image`. Reason with
the visual information step by step, and put your final answer within \\boxed{}.
```

## E.2 Prompts Used in Data Synthesis

### E.2.1 Prompt for Question-answer Pair Generation for SA1B

Since SA1B lacks question-answer pairs and corrisponding annotations and some pictures in SA1B
have little content, we prompt gpt-4o to first determine if the image is information-rich. If yes, gpt-4o
needs to use zoom-in tool to first crop a small part of the image, and then ask a question about objects
in the small region. Otherwise, gpt-4o should reply Not valid. Here is the prompt for gpt-4o:

```
You are an expert in generating questions about small details in a
image. You will be given a HR image. First determin if the image is an
information-rich image. If it is not, return 'Not valid'. If it is,
choose a small region and use crop image tool to zoom in. According to
both cropped image and whole image. Generate a question about objects in
the small region. The question should be about the small object or its
color, material. Also generate 4 choices. One of them is the correct
answer. Others are wrong. It should not be ambiguous. For example if you
ask about the color of a person's shoes, there should either be only one
person or you specify which person you are referring to. Please make
sure the object is small. Don't ask about questions related to the
cropped image. For example, don't ask 'What is the color of the frame in
the cropped image?' because the cropped image will not be provided. Put
the question in the following format:
<question>
QUESTION HERE
</question>
Here is an example question:
<question>
question:What is the color of the person's shoes?
choices:
A: Red
B: Blue
C: Green
D: Yellow
correct_answer: A
</question>
<question>
question:What is the child on the crosswalk holding?
choices:
A: Ice cream
B: Ball
C: Book
D: None
correct_answer: C
</question>
Here is the tool description {tool_description}. For each tool call,
return a json object with function name and arguments within
<tool_call></tool_call> XML tags:
<tool_call>
{{"name": <function-name>, "arguments": <args-json-object>}}
</tool_call>
Stop generating after you call a tool.
Here is the image.
```

### E.2.2 Prompts for Expert Trajectory Synthesis

**For SA1B dataset:**

```
You are an expert in generating trajectories involving image cropping and answering
```

questions. You will be given an image and one cropped part of it and a question. First,
you need to briefly analyze the whole image, then generate: "Now I will zoom in to look
clearer at 'query object or text'." Then you need to analyze the cropped part and answer
the question. Put your answer choice in \boxed{}.

Here is an example:

question: What is the price mentioned for renting the single house?
choices:
A: 9,000 Baht
B: 10,000 Baht
C: 8,500 Baht
D: 12,000 Baht

Analyzing the whole image: The image shows a lively street scene with people
celebrating, possibly during a festival. There is a pickup truck with people
on it, and others walking around. A signboard with text is visible in the
background, which seems to contain information about renting or selling a house.

Now I will zoom in to look clearer at the text on the signboard.

Analyzing the cropped part: The cropped image focuses on the signboard. The text on the
signboard mentions "SALE / RENT SINGLE HOUSE" and specifies the price for renting as
**9,000 Baht**.

\boxed{A}

Here is the question, image and cropped part:
{text}

**For Fineweb dataset:**

You are an expert in generating trajectories involving image cropping and answering
questions. You will be given an image and one cropped part of it and a question.
First you need to briefly analyze the whole image, then generate: "Now I will zoom
in to look clearer at the part about 'query'." Then you need to analyze the cropped
part and answer the question. Put your answer in \boxed{}. Final answer should be
text from article. Don't change the original text or include irrelevant text from the
article. The answer should be in one sentence.

Here are some examples:

question: What are the key responsibilities of a leader?

Analyzing the whole image: The document appears to be an article titled "Top 7
Skills a Leadership Training Should Teach Managers." It discusses various aspects
of leadership training, including leadership essentials, change management,
performance coaching, and conflict management. The article emphasizes the
importance of leadership skills in managing teams effectively.

Now I will zoom in to look clearer at the part about "key responsibilities of a
leader."

Analyzing the cropped part: The cropped part focuses on "Leadership Essentials,"
which outlines the basics of leadership, including understanding the role of a
leader and the key responsibilities of a leader.

\boxed{building relationships, setting expectations, delegation, and developing
a goal-oriented approach.}

question: Who won the first SEC championship in football?

Analyzing the whole image: The document is a Wikipedia article titled "SEC
Championship Game." It provides an overview of the Southeastern Conference (SEC)
Football Championship Game, including its history, format, results, and notable

```
moments. The article also includes a table summarizing the results of all SEC
Championship games since its inception in 1992.

Now I will zoom in to look clearer at the part about "who won the first SEC
championship in football."

Analyzing the cropped part: The cropped section includes a table of results from
all SEC Championship games. The first game, held in 1992, lists #2 Alabama
defeating #12 Florida with a score of 28-21 at Legion Field in Birmingham, Alabama.

\boxed{Alabama}

Here is the question and image:
{text}
```

**For STARQA dataset:**

```
You are an expert in generating trajectories involving frame selection and answering
questions. You will be given 16 images (video frames) in chronological order and
several selected frames from them and a question. First you need to briefly analyze
the whole video, then generate: "Now I will select some frames to look clearer at
'query object or text'." Then you need to analyze the selected frames and answer the
question. Put your answer choice in \boxed{}.

Here are some examples:

question: why did the woman take the measuring spoons away from the boy?
choices:
A: do not need it anymore
B: feeding
C: finish eating the piece
D: so can take picture
E: wants to play with it

Analyzing the video:
The video shows a woman and a boy in a kitchen setting. The boy is sitting on the
counter, holding measuring spoons, while the woman appears to be engaged in a baking
or cooking activity. The woman interacts with the boy, guiding him as they work with
ingredients like flour and eggs. Toward the end, the woman takes the measuring spoons
away from the boy.

Now I will select some frames to look clearer at why the woman took the measuring
spoons away from the boy.

Analyzing the selected frames:
In the selected frames, the woman is seen taking the measuring spoons from the boy.
The boy appears to have finished using the spoons to add ingredients to the bowl.
The woman likely takes the spoons to proceed with the next step in the cooking process.

\boxed{A}

question: Which object was put down by the person?
choices:
A: The cup/glass/bottle.
B: The clothes.
C: The bag.
D: The book

Analyzing the video:
The video shows a person entering a room and sitting at a table. The person appears
to be holding a sandwich and a book. She places the book on the table, eats the
sandwich, and then picks up the book again to read. Toward the end of the video, the
person leaves the table, leaving the book behind.

Now I will select some frames to look clearer at which object was put down by the
```

person.

Analyzing the selected frames:
In the selected frames, the person is seen entering the room holding a sandwich and a book. She places the book on the table before eating the sandwich. The book remains on the table as the person continues her activity and eventually leaves the room.

\boxed{D}

Here is the question and video:
{text}

