# OpenReview forum: "Pixel Reasoner: Incentivizing Pixel Space Reasoning via Curiosity-Driven Reinforcement Learning"
_NeurIPS.cc/2025/Conference — NeurIPS 2025 poster_

### Official Review · Reviewer_jxhz · 2025-07-02

**Clarity:** 3
**Significance:** 2
**Originality:** 2
**Rating:** 4
**Confidence:** 3

**Summary:**

This paper introduces the concept of pixel-space reasoning, a novel paradigm for Vision-Language Models (VLMs) that enables direct interaction with visual inputs through operations such as zoom-in and select-frame. Unlike traditional chain-of-thought reasoning confined to textual space, this approach allows VLMs to inspect and manipulate visual content directly during the reasoning process, improving their ability to handle visually intensive tasks. To overcome challenges in training VLMs to effectively use these new capabilities—such as imbalanced proficiency between text-based and visual reasoning—the authors propose a two-phase training framework: (1) warm-start instruction tuning using synthesized reasoning traces, followed by (2) curiosity-driven reinforcement learning (RL) that incentivizes exploration of pixel-space reasoning while maintaining efficiency.

**Questions:**

1. Generalization Beyond Zoom-In and Select-Frame

The paper demonstrates success using zoom-in and select-frame operations. How would the framework perform if extended to include additional visual operations (e.g., drawing, color filtering, object highlighting)? Are there plans or preliminary results indicating the scalability of the method to more complex visual tools?

2. Error Analysis and Failure Cases

The paper includes some analysis of failure rates in visual operation execution. Could you provide concrete examples of cases where the model failed to utilize pixel-space reasoning effectively, and what insights these failures offer for future improvements?

**Ethical Concerns:**

["NO or VERY MINOR ethics concerns only"]

**Final Justification:**

borderline accept

**Limitations:**

The authors have adequately addressed the limitations of their work. They acknowledge the current restriction to two visual operations and suggest that the framework is easily extensible. They also discuss the need for further research into richer visual operations and broader task coverage. However, the paper could benefit from a more detailed discussion of failure cases.

**Paper Formatting Concerns:**

No obvious formatting issues.

**Quality:**

2

**Strengths And Weaknesses:**

## Strengths
- Originality

The core idea of pixel-space reasoning is novel and addresses a clear gap in current VLMs—namely, their reliance on textual reasoning even when visual information is critical.

- Clarity
The paper is well-written and structured, making it easy to follow the problem formulation, methodology, and results.
Figures and examples are used effectively to illustrate the reasoning process and training dynamics.

## Weaknesses:

- Limited Exploration of Visual Operations

The study currently focuses only on zoom-in and select-frame operations. While these are useful, broader visual manipulations (e.g., drawing, masking, depth estimation) could further enhance the model's capabilities and generalization.

---

> ### Author Rebuttal · Authors · 2025-07-29
>
> We thank you for your thoughtful questions and constructive feedback. Below, we address the questions.
>
> **1. On Generalization Beyond Zoom-In and Select-Frame**
>
> We thank the reviewer for requesting us for clarification on this crucial question. We agree that the current set of operations is focused, a deliberate choice we acknowledge in the paper's limitations section. We would like to clarify our rationale.
>
> 1.  **Rationale for a Focused Scope:** The space of potential visual operations is vast, including rotation, drawing, filtering, segmentation, and more. Rather than attempting an exhaustive implementation, which is neither feasible nor necessarily productive for an initial investigation, we strategically selected **ZOOM-IN** and **SELECT-FRAME**. These were chosen because they are fundamental primitives for interacting with static images and dynamic videos, respectively. This focused set allowed us to rigorously study the core challenges of the pixel-space reasoning paradigm in a controlled yet broadly representative setting. These two operations cover the most of the use cases. The other operations like segmentation, etc are not contributing much values to our primary goal, i.e., to first establish the importance and viability of this paradigm.
>
> 2.  **Extensibility of the Framework:** Our framework is designed from the ground up to be extensible. We envision future work building upon this foundation by designing bespoke operations tailored to specific downstream tasks (e.g., line drawing for geometric problem-solving, or fine-grained segmentation for autonomous driving). The core mechanisms for incorporating new operations would remain consistent:
>     * **Skill Acquisition via Instruction Tuning:** The model would be fine-tuned on instruction-following datasets to learn the syntax and semantics of any new visual operation.
>     * **Exploration via Curiosity:** The curiosity-based reward mechanism is agnostic to the specific operation and would continue to encourage the model to explore the utility of the new tool.
>     * **Task-Specific Rewards:** The primary adaptation would be in the task-success reward function. For example, extending to a GUI grounding task would involve incorporating an Intersection-over-Union ($IoU$) reward, while a document understanding task might require a novel layout-aware reward metric.
>
> In summary, while the scope of the *demonstrated* operations is focused, this does not detract from the paper's main contribution: revealing the importance of pixel-space reasoning and providing a viable, scalable method to unlock this capability in VLMs.
>
>
> **2. On Error Analysis and Failure Cases**
>
> We appreciate the reviewer's request for a more detailed discussion of failure cases. As noted, we provide a qualitative analysis in Appendix D.2. The primary failure modes observed were:
>
> 1.  **Hallucination:** In some cases, the model pretends the visual operation is performed successfully and hallucinate with a conclusion and continue its reasoning.
>
> 2.  **No-Reaction:** Another failure mode is the model's inability to properly react to the errors from visual operations. The model noted the errors and then proceed with textual reasoning without attempting to fix the errors.
>
> **Insights and Future Work:**
> This paper is a preliminary study of pixel-space reasoning and it follows the common setting of using only outcome rewards. A key insight from analyzing these failures is that **intermediate supervision** could benefit more accurate and reliable pixel-space reasoning. Currently, the model relies primarily on the outcome reward, so the correctness of intermediate steps, e.g., correcting errors from visual operations, will not be properly encouraged. This points to a clear path for future improvement: developing methods to provide denser supervisory signals for the intermediate pixel-space reasoning steps. This could involve designing reward shaping for handling errors from visual operations, or providing more signals for the correctness of visual operations, e.g., rewarding the grounding of a relevant visual cues.

---

> > ### Comment · Reviewer_jxhz · 2025-08-06
> >
> > Thank the author for the clear and thoughtful explanation. The response addressed my question thoroughly and provided valuable insight. I truly appreciate the effort and clarity. After careful consideration, I have decided to maintain my original score.

---

> > > ### Author Response · Authors · 2025-08-06
> > >
> > > Thank you once again for the thought-provoking suggestions, they have helped enhanced our paper and both the analysis and discussion on extending visual operations will be included in our revision.

---

### Official Review · Reviewer_RbYh · 2025-07-02

**Clarity:** 3
**Significance:** 3
**Originality:** 3
**Rating:** 4
**Confidence:** 4

**Summary:**

This paper proposes an interesting idea: leveraging pixel-space reasoning (via visual operations) to improve visual reasoning capabilities when tackling complex visual understanding tasks such as InfographicsVQA. The approach is concise and well-motivated. However, several important technical details are missing, making it hard to follow. The authors are encouraged to improve the writing and evaluate the proposed method on more challenging benchmarks—such as ChartQAPro, which emphasizes visual reasoning.

**Questions:**

Please carefully address the concerns raised in the “Weaknesses” section above.

**Ethical Concerns:**

["NO or VERY MINOR ethics concerns only"]

**Final Justification:**

The authors fail to address the raised major concerns, such as limited experimental results.

**Quality:**

3

**Strengths And Weaknesses:**

**Strengths:**

* The idea of pixel-level reasoning via a set of image operations is reasonable and appears to be inspired by a recent blog post from OpenAI \[1]. The authors should cite this blog and credit the original source accordingly.

* The proposed synthetic dataset also represents a valid contribution to the research community.

\[1] [https://openai.com/index/thinking-with-images/](https://openai.com/index/thinking-with-images/)

---

**Weaknesses:**

1. **Limited experimental results:**
   The primary concern is that the authors fail to demonstrate the effectiveness of the proposed approach on tasks requiring highly accurate visual reasoning, such as the ChartQAPro benchmark.

   \[1] *ChartQAPro: A More Diverse and Challenging Benchmark for Chart Question Answering*, arXiv:2504.05506

2. **Lack of discussion on the limitations of the synthetic dataset:**
   The reviewer is also concerned about how the authors plan to construct a scalable visual reasoning dataset for such a challenging task.

3. **Insufficient analysis of the 'learning trap':**
   As one of the core contributions of the paper, the concept of a 'learning trap' lacks systematic theoretical analysis and deeper discussion.

4. **Concerns regarding ablation studies:**
   The ablation results show that the error-correcting data significantly improves performance. However, it remains unclear why this data enhances the model’s visual reasoning or tool invocation capabilities. Further analysis is strongly encouraged.

5. **Advantages of using RL for pixel reasoning**. The necessity or reason for using RL should be further explained. Does the model's pixel-space reasoning capability necessarily require learning through RL, or is RL a more efficient approach?

---

> ### Author Rebuttal · Authors · 2025-07-29
>
> We sincerely thank the reviewer for constructive suggestions. Below, we address the reviewer's specific questions.
>
> **1. Regarding the inclusion of the ChartQAPro benchmark.**
>
> We thank the reviewer for bringing the interesting ChartQAPro benchmark to our attention. We agree that evaluating our method on such tasks is a valuable direction. However, we did not include it in the current submission for two primary reasons:
>
> * **Timing:** The ChartQAPro benchmark was released in mid-April 2025. Given the NeurIPS submission deadline in May, we didn't know about this new work at the moment of submission.
> * **Scope:** As we state in our paper (L34, L85), our primary research goal is to demonstrate the significance of pixel-space reasoning in visually-intensive, general-purpose scenarios. We therefore prioritized benchmarks like InfographicsVQA, V-Star, MVBench, which include visually-rich scenarios like videos, high-resolution scenes and infographics.
>
> **2. Regarding the limitations of the synthetic dataset.**
>
> We appreciate the reviewer's insightful comment on the limitations and scalability of our synthetic dataset. We acknowledge that its current form is focused on demonstrating the core capabilities on visually-rich tasks and does not yet cover complex, multi-step numerical or logical reasoning often required by chart-based VQA.
>
> To address this and to outline a path toward a more comprehensive dataset, we will add a discussion on this limitation to the paper. We will also propose a concrete pipeline for extending our synthetic data generation to tasks like ChartQA, which includes the following steps:
> 1.  Leverage a state-of-the-art VLM (e.g., GPT-4o) to identify key visual elements and tabular data within a chart.
> 2.  Generate question-answer pairs that require targeted reasoning about these identified visual cues.
> 3.  Synthesize both expert demonstration and self-correction trajectories based on these generated pairs, teaching the model to ground its reasoning in specific visual regions.
>
> **3. Regarding the analysis of the 'learning trap'.**
>
> We thank the reviewer for highlighting the need for a deeper analysis on the 'learning trap'.
>
> In our current work, we provided a deep empirical analysis of this "learning trap" problem in Section 5.2. Our experiments demonstrate that without instruction tuning and intrinsic rewards, the model defaults to its well-established textual reasoning abilities and fails to effectively learn the novel pixel-space reasoning skill, which is precisely the "learning trap" problem. We also show that the fundamental reason for this problem is the significant proficiency gap between the two capabilities.
>
> To make this clearer, we will revise the manuscript to explicitly frame Section 5.2 as an empirical investigation into the 'learning trap'.
>
> **4. Regarding the ablation study on self-correction data.**
>
> We thank the reviewer for raising this point, as it gives us the opportunity to clarify the precise role of the self-correction data. We believe the potential confusion arises from our choice to use a negative framing in the paper (L302). To emphasize the data's impact, we described the model’s failures without it, rather than its benefits with it.
>
> To state the mechanism directly and positively: the self-correction trajectories are designed to teach the model to handle errors from the visual operations. When a visual operation yields an unexpected or erroneous outcome, this data provides explicit demonstrations of how to proceed with a corrective action, rather than leading to model's bypassing of these errors from visual operations.
>
> In our revision, we will rephrase this section to state this positive mechanism more explicitly, ensuring the contribution of the self-correction data is unambiguous.
>
> **5. Regarding the advantages of using RL for pixel-space reasoning.**
>
> We thank the reviewer for requesting a more thorough justification for using Reinforcement Learning (RL). As we note in L52, our approach follows the standard paradigm of post-training with instruction tuning and RL to instill new, complex abilities. The reason this two-stage process is necessary is that instruction tuning alone is insufficient for this task.
>
> As our results in Table 1 empirically demonstrate, a model trained with only instruction tuning is fragile and struggles to generalize. RL is not just a means of making learning more efficient; it is *necessary* for developing a robust reasoning policy. RL enables the model to explore the vast space of possible visual operations and learn to assign credit to actions over long sequences. The intrinsic reward signal encourages the model to move beyond simply mimicking expert demonstrations and to discover for itself which policies lead to correct answers, making it more resilient to novel situations not seen during supervised fine-tuning. We will clarify this motivation in the revised manuscript.

---

> ### Comment · Reviewer_RbYh · 2025-08-04
> **The authors do not address the major concerns.**
>
> Thank you to the authors for the concise response. However, I did not find any informative response to the raised major concerns,**such as limited experimental results**.

---

> ### Author Response · Authors · 2025-08-04
> **New Results on ChartQAPro Benchmark**
>
> **We have now completed our experiments on CharQAPro Bechmark and list our results below.**
>
> We compare PixelReasoner-7B against its base model (Qwen-VL2.5-7B), and other leading models reported in the ChartQAPro paper. For PixelReasoner-7B, the values in parentheses next to each score indicate the rate of its pixel-space reasoning.
> | Model                | Factoid       | MCQ            | Convers.       | FactChk.       | Hypoth.       | Overall            |
> |----------------------|---------------|----------------|----------------|----------------|---------------|--------------------|
> | GPT4o                |          37.4 |          61.68 |          33.93 |          57.37 |         30.83 |              41.68 |
> | Gemini-Flash-2.0     |         51.51 |          69.15 |          43.84 |          67.62 |         39.89 |              53.66 |
> | Gemini-Flash-1.5     |         42.37 |          64.01 |          40.17 |          56.14 |         39.42 |              45.97 |
> | Claude Sonnet 3.5    |         53.61 |          78.03 |          43.84 |          65.16 |         46.11 |          **55.81** |
> | Intern-VL2.5-1B      |          5.45 |           0.46 |          14.86 |          21.17 |         17.08 |               8.96 |
> | Intern-VL2.5-2B      |          9.42 |           6.07 |          13.02 |          36.06 |         19.23 |              13.46 |
> | DeepSeek-VL2-3.4B    |          9.63 |            1.4 |          18.09 |          38.11 |         23.25 |              14.33 |
> | Phi 3.5-Vision-4B    |         10.55 |          32.71 |           27.2 |           8.19 |          8.16 |              15.23 |
> | Qwen-VL2-7B          |         32.95 |          46.26 |           37.6 |           50.4 |         29.65 |              37.17 |
> | Intern-VL2.5-8B      |         29.53 |          23.36 |          28.87 |          56.14 |         27.73 |              31.99 |
> | LLaMA 3.2-Vision-11B |         19.65 |          47.66 |          19.15 |          44.45 |          13.1 |              25.43 |
> | Qwen-VL2.5-7B        |         37.28 |          66.12 |          36.47 |          58.19 |         45.82 |              43.27 |
> | PixelReasoner-7B     | 40.82 (14.2%) | 67.75 (11.21%) | 44.45 (19.74%) | 67.62 (10.24%) | 45.92 (10.2%) | **47.97 (15.02%)** |
>
> Our analysis highlights the following key findings:
>
> - **State-of-the-Art Open-Source Performance:** PixelReasoner-7B establishes a new state-of-the-art among open-source models with an overall score of 47.97. It outperforms its base model, Qwen-VL2.5-7B, by 4.7 absolute points on the overall score.
>
> - **Generalization:** This strong performance is achieved without any fine-tuning on chart-related datasets. This result underscores the excellent generalization capability of our model's inherent pixel-space reasoning mechanism for a complex task it was not explicitly trained for.
>
> - **Computational Efficiency:** These performance gains are realized without introducing significant computational overhead. The overall RaPR is 15%, making our approach both effective and highly efficient compared to the base model.
>
> - **Competitive with Proprietary Models:** While Claude 3.5 Sonnet leads overall, our open-source model is highly competitive, surpassing other powerful proprietary models like GPT-4o (41.68) and Gemini 1.5-Flash (45.97).
>
> - **Limitation on Unanswerable Questions:** A detailed error analysis revealed a key limitation: PixelReasoner-7B currently fails on most unanswerable questions within the benchmark. Addressing this failure mode is a clear priority and a promising direction for future work.
>
> Hope our new experimental results can resolve your concerns regarding our submission.

---

> > ### Comment · Reviewer_RbYh · 2025-08-04
> > **Thank you for the prompt response.**
> >
> > Since the authors have thoroughly addressed the major concerns, I will maintain my original positive rating, provided that they release the source code for reproduction.

---

> > > ### Author Response · Authors · 2025-08-05
> > >
> > > Thank you once again for bringing the exciting benchmark to our attention. We will include the results in our revision. We also note that code,data,models were released but the policy disallow us to disclose the link, so we will update upon paper acceptance. We will make sure this release include the evaluation results on ChartQAPro.

---

### Official Review · Reviewer_1tA7 · 2025-07-03

**Clarity:** 4
**Significance:** 4
**Originality:** 3
**Rating:** 6
**Confidence:** 4

**Summary:**

This paper presents pixel reasoner, a framework to train multimodal LLM to do complex multimodal reasoning with two tools: zoom-in and select-frame. The training recipe contains two stages, SFT and RL. Specifically, the authors tackle numerous challenges they face and propose novel and effective solutions for each stage, including the warm-start instruction tuning and the curiosity-driven RL. The model achieves great results on V*, TallyQA, MVBench, and InfographicVQA, outperforming all other open-source models. Extensive ablation experiments show the effectiveness of each step in the training recipe.

**Questions:**

1. What is the number of tokens & tool-calls per query for each task? It is interesting to see these methods' efficiency trade-offs.

2. What other operations besides zoom-in and select-frame might be useful for multimodal reasoning? How would you change the algorithm if there are more tools?

**Ethical Concerns:**

["NO or VERY MINOR ethics concerns only"]

**Limitations:**

yes

**Paper Formatting Concerns:**

No concern.

**Quality:**

4

**Strengths And Weaknesses:**

Strength:
1. The motivation is clear, and the method is effective.
2. The meticulous instruction tuning stage and the RL stage are very informative to other researchers. The paper is well-written and clearly shows the importance of these tricks and the mathematical intuition behind them.
3. SOTA performance on a wide range of challenging tasks.

Weakness:
In Table 1, the VPD model numbers are for TallyQA, not V*. Also, they are not tool-use models (they are trained with tool-use trajectories). The authors should correct these.

---

> ### Author Rebuttal · Authors · 2025-07-30
>
> We thank the reviewer for their valuable feedback and insightful questions.
>
>
> **1. Efficiency Trade-offs (Tokens & Visual Operations)**
>
> Thank you for this excellent question about the efficiency of our method. We reported RaPR (rate of pixel-space reasoning) and tokens per query-response for each benchmark to clarify the computational trade-offs.
>
> | Benchmark | RaPR | Avg. Tokens |
> | :--- | :--- | :--- |
> | V* (Visual Search) | 0.78 (150/191) |  153.8 |
> | TallyQA (Visual Counting) | 0.43 (1389/3200) | 151.2 |
> | MV-Bench (Video Reasoning) | 0.94 (3802/4032) |  215.4 |
> | InfographicsVQA (Doc Reasoning) | 0.21 (593/2816) |  161.5 |
>
> As shown, tasks requiring fine-grained detail, like video analysis (MV-Bench) and visual search (V*), trigger visual operations most frequently. In contrast, tasks like InfographicsVQA, where the evidence is often easier to found in the single page infographics, have a much lower rate of visual opeartion. These observations also demonstrate PixelReasoner's ability to allocate computation efficiently and autonomously.
>
> **2. Extending with More Visual Operations**
>
> This is a great question about the extensibility of our framework. Additional operations like **line-drawing** for geometric problems, **region highlighting/masking** for grounded reasoning, or **optical character recognition (OCR)** for document understanding would be highly beneficial. Our framework is designed to be easily extensible to new tools through three core mechanisms:
>
> * **Skill Acquisition:** New operations are incorporated by fine-tuning the model on instruction-following datasets that demonstrate the tool's syntax and function.
> * **Agnostic Exploration:** Our curiosity-driven reward mechanism is tool-agnostic. It naturally encourages the model to explore when and how a new tool can be used to solve a task, without needing manual reprogramming.
> * **Task-Specific Rewards:** The primary adaptation is in the task-success reward function. For example, a GUI navigation task could use an **Intersection-over-Union (IoU)** reward for object grounding, while a document task might use a **layout-aware edit distance**. This modular design allows our method to easily incorporate new operations for a wide range of multimodal problems.
>
>
> **3. Correction in Table 1**
> We sincerely thank the reviewer for identifying the error in Table 1 and the imprecise description of the VPD model. We will correct them accordingly in our revision.

---

### Official Review · Reviewer_wysE · 2025-07-03

**Clarity:** 2
**Significance:** 3
**Originality:** 3
**Rating:** 5
**Confidence:** 4

**Summary:**

The paper introduces Pixel Reasoner, a novel VLM framework that enables pixel-space reasoning, meaning the model actively performs visual operations (like zooming in or selecting frames) as part of its reasoning process, rather than relying exclusively on CoT reasoning. The proposed method first synthesizes expert visual reasoning trajectories, including self-correction paths, to teach the model how and when to use visual operations effectively. Then, curiosity-driven RL employs a carefully designed reward structure to overcome the inherent bias of the model toward text-based reasoning. The reward encourages the use of visual operations while penalizing excessive or redundant operations, mitigating what the paper term a "learning trap", i.e., the model’s tendency to default to textual reasoning because it is initially more proficient.

**Questions:**

See weaknesses above. Not sure whether this is non-trivial, but could the authors provide more analysis or statistics on how often pixel-space reasoning is truly necessary versus when it is artificially invoked to meet the rate constraint? Or perhaps check what happens in examples from existing benchmarks in which pixel reasoning is not necessary?

**Ethical Concerns:**

["NO or VERY MINOR ethics concerns only"]

**Final Justification:**

The rebuttal satisfactorily addressed all major concerns. The authors clarified the reasoning behind the selected set of visual operations, provided a detailed explanation of the RaPR (dynamic) mechanism, and resolved all concerns regarding baselines and data quality. My final recommendation has been updated to reflect the novelty and significance of operationalizing pixel-space reasoning, supported by the clear author responses.

**Limitations:**

A limitations section exists, but I could not find a broader impacts section.

**Paper Formatting Concerns:**

No issues

**Quality:**

3

**Strengths And Weaknesses:**

Strengths:
- The concept of pixel-space reasoning is a significant and timely contribution that opens new directions in VLM reasoning.
- The paper is methodologically rigorous and combines instruction tuning with curiosity-driven reinforcement learning.
- The experiments are comprehensive, covering multiple benchmarks, and the performance gains against Qwen2.5-VL-7B are consistent. The demonstrated performance, especially surpassing most of the open-source and proprietary models, indicates that the proposed methods meaningfully improve practical VLM capabilities.

Weaknesses:
- The scope of visual operations is currently narrow. While the framework is extensible, the demonstrated significance is somewhat confined to the selected benchmarks and the two operations presented.
- Some components, such as curiosity-driven exploration and instruction tuning, are adapted from existing RL literature and VLM fine-tuning pipelines with no clear technical contributions, though the paper clearly motivates these design choices.
- The SA1B dataset, while large and visually diverse, is inherently noisy because it is generated through automatic segmentation at scale. For example, it is not clear whether the fine-grained visual cues used in the instruction tuning phase are consistently reliable. Although the work attempts to mitigate this by filtering for high-information regions using GPT-4o, this can similarly introduce biases or systematically exclude challenging cases. There is a risk that the model might learn brittle or superficial strategies with limited generalization.
- Could the fixed-rate constraint on pixel-space reasoning (RaPR) lead to over-regularization? If there exist queries that do not require pixel-space reasoning at all, yet the constraint enforces a minimum rate across the dataset, then the model may superficially boost RaPR to meet the target without genuinely improving reasoning quality, e.g., by adding visual operations even when they are irrelevant.
- Will the data be released? The submission did not contain any data samples for reviewers to assess data quality.
- In Table 1, some cells have no reported numbers. While it is reasonable in some cases, the Models with Tools have results for only one benchmark, which makes it challenging to evaluate the performance gains against these baselines. Also, SEAL [Wu and Xie, 2024] h is not included as a baseline, and related work does not explain the differences of this work with existing visual search models.
- The absence of qualitative examples in the main paper is striking. Given the paper’s focus on a new reasoning modality, it would be valuable to see more concrete reasoning trajectories that illustrate when pixel-space reasoning is essential, when it fails, and when it might be unnecessarily invoked.

Comments:
- Figure 3 fonts are too small. An actual, fully developed example would help understand the dataset better. Figure 8 is also unreadable.
- Figure 7 highlights that pixel-space reasoning does not naturally emerge as a dominant strategy without curiosity-driven incentives. This suggests that the pixel-space reasoning capability is not intrinsically self-reinforcing. This raises an intriguing question: can we develop more intrinsic mechanisms, where the model autonomously values pixel-space operations based on their long-term utility or epistemic uncertainty, rather than needing hard-coded curiosity bonuses? Such directions could potentially lead to more robust, adaptive, and agent-like reasoning behaviors in future vision-language models.

---

> ### Author Rebuttal · Authors · 2025-07-29
>
> We sincerely thank the reviewer for their detailed, insightful, and constructive feedback. We are encouraged that they recognized our work as a "significant and timely contribution" and appreciated the rigor of our methodology and the meaningful performance gains our model achieves.
>
> We would like to begin by highlighting what we see as the core contribution of this work: to formally identify, operationalize, and incentivize pixel-space reasoning. We believe this is a distinctive capability that underpins the powerful visual analysis seen in frontier models like OpenAI o3. Our paper reveals a key challenge in acquiring this new ability and proposes an effective framework to overcome it. We will frame our responses to the specific points below with this central contribution in mind.
>
>
> ### **Addressing Weaknesses**
>
> **1. On the Scope of Visual Operations:**
> We agree with the reviewer that the current set of visual operations is focused, a point we also acknowledged in the limitation section. This was a deliberate choice for this initial work, and we wish to clarify our rationale:
> - The space of potential visual operations is vast (e.g., rotation, line drawing, segmentation, and more). It is neither feasible nor necessarily productive to try and enumerate and implement them all.
> - Rather than attempting an exhaustive implementation, we selected ZOOM-IN and SELECT-FRAME as they are fundamental and broadly representative operations for interacting with static images and dynamic videos, respectively. These two operations cover the most of the use cases. The other operations like segmentation, etc are not contributing much values to our primary goal, i.e., to show the importance of the pixel-space reasoning paradigm.
> - Our framework is designed to be extensible. Based on the current core set, we envision future work designing bespoke operations for specific downstream tasks, such as line drawing for geometric problem-solving or fine-grained segmentation for autonomous driving, etc.
>
> Therefore, while the scope of the demonstrated operations is focused, we believe this does not detract from the paper's main contribution: revealing the importance and challenges of pixel-space reasoning and providing a viable method to unlock this capability in VLMs.
>
> **2. On the Technical Contribution of the Training Pipeline:**
> We appreciate the reviewer's point that we indeed "stand on the shoulders of giants". We will refine the contribution stated in line 83 by clarifying that we innovatively address the challenges of learning pixel-space reasoning by proposing the approach, but not directly inventing instruction-tuning from scratch.
>
> **3. On the Quality of the SA1B Dataset and Potential for Brittle Strategies:**
> We thank the reviewer for raising this important point about data quality. We share the concern regarding the potential noise in automatically generated datasets and took several deliberate steps to mitigate this risk, which we would like to clarify:
> - **Diverse Data Sources:** We did not rely solely on SA1B. Our instruction tuning data is curated from a variety of sources, including FineWeb and STARQA, to provide rich textual and contextual diversity (Lines 117-119).
> - **Controlled Synthesis:** Our data generation process utilizes a template-based synthesis approach (Figure 3, Lines 137-141). This provides strong structural control over the reasoning traces, ensuring that the generated visual operations are logically sound and contextually necessary for the task. This moves beyond simple noisy labels and enforces a coherent reasoning structure.
> - **Empirical Generalization:** The strongest evidence against the model learning brittle strategies lies in its performance. Our model demonstrates consistent and significant generalization across a wide range of benchmarks and media formats (images and videos), many of which have distributions distinct from the training data.
>
> These factors together ensure that the model learns robust and generalizable reasoning skills rather than superficial shortcuts. We thank the reviewer for highlighting this concern, and we will consider more measures to facilitate data quality assurance.
>
> **4. On the Fixed-Rate Constraint (RaPR) and Potential Over-Regularization:**
> This is a crucial question, and we thank the reviewer for giving us the opportunity to clarify a nuanced but central aspect of our method. There may be a misunderstanding that the `RaPR` constraint is a rigid, dataset-wide mandate. In fact, it is a dynamic reward bonus to that monitors at the query level and rewards at the response level.
>
> Here is a more precise breakdown of how it functions:
> - RaPR monitors the query-level exploration. It tracks the proportion of rollouts that uses visual operations within the same query group.
> - A curiosity bonus is added to a response's reward if and only if two conditions are met: (1) the response contains a visual operation, and (2) the current query-level RaPR is too low, i.e., below the target ratio.
> - **The model is never forced to use a visual operation.** On the one hand, if the model determines that visual operations are not beneficial for a query and consequently explores no such actions in its rollouts, no penalty is incurred. On the other hand, if the model finds visual operations useful occasionally and its exploration satisfies the curiosity, no further reward bonus is given, thus preventing the model from adding gratuitous operations.
>
> - The curiosity bonus acts as a temporary learning scaffold, primarily to help the model escape the initial "learning trap." As Figure 6 shows, this reward signal diminishes as the model learns the intrinsic value of the actions. The primary reward for reasoning correctness always remains dominant.
>
> Therefore, there is no permanent mandate that could lead to over-regularization. We will revise the methodology section to make this dynamic, query-level exploration mechanism clearer to prevent misunderstanding.
>
>
> **5. On Data Release:**
> We are fully committed to open science. The data,code,models were released (as stated in Line 253), but rebuttal disallow any form of link sharing. So we will share the link after the paper decision is out on the Openreview platform. We also appreciate the suggestion to include data samples and will add representative examples to the appendix in our revision.
>
> **6. On Table 1 and Missing Baselines:**
> - **SEAL Baseline:** We thank the reviewer for this point and would like to gently clarify that SEAL is included in Table 1 and cited in our related work (Line 355). To improve clarity, we will explicitly name "SEAL" in our discussion of "visual guided search" in Line 355 of the related work section.
> - **Missing Tool-Use Model Results:** The empty cells for some tool-using models exist because the original papers for these models did not report performance on those specific benchmarks. Adapting these highly specialized visual search models to the diverse reasoning benchmarks we use is a significant engineering effort beyond the scope of this work. We believe the current table provides a fair comparison on the visual search benchmark (V-Star) where results are available.
> - **Distinction from tool-using models:** Our work focuses on enabling a new reasoning paradigm where the model directly interacts with visual inputs to guide its internal thought process. This is distinct from direct tool-use or visual search, where the goal is typically to retrieve external information. The objective of our visual operations is to improve the quality of reasoning itself. We will clarify this distinction further in the related work section.
>
>
> **7. On the Absence of Qualitative Examples in the main paper:**
> We agree that qualitative examples are essential for a paper introducing a new reasoning ability. Due to page constraints, we placed these examples, including both successes and failures, in the appendix. In our revision, we will restructure the paper to move several key qualitative examples into the main body for better visibility.
>
> ### **Addressing Comments and Questions**
>
> * **Figure Readability:** We agree and will revise Figures 3 and 8 for the final version to improve readability.
>
> * **Intrinsic Mechanisms for Pixel-Space Reasoning:** We are grateful for this insightful and thought-provoking question. The reviewer's observation that pixel-space reasoning is not "intrinsically self-reinforcing" perfectly captures the motivation behind our work. This is precisely the "learning trap" we aimed to solve. Our use of curiosity-driven RL is a pragmatic and effective method to provide an initial "intrinsic motivation" and bootstrap the learning process, allowing the model to discover the long-term utility of these actions.
>
> We wholeheartedly agree that developing more sophisticated intrinsic mechanisms is an exciting and important frontier for research. We believe our work is a crucial first step in this direction. We will add a discussion of this promising future direction to our conclusion to inspire further research in the community.
>
> * **Broader Impacts Section:** We apologize for the confusion. A broader impacts statement was included in the NeurIPS checklist (Lines 703-710). We will add a formal "Broader Impacts" section to the appendix for better visibility.
>
> We thank the reviewer again for their time and their thoughtful, high-quality feedback. This review has provided us with clear, actionable guidance that will undoubtedly help us improve the quality and clarity of our paper.

---

> > ### Comment · Reviewer_wysE · 2025-08-01
> > **Thank you**
> >
> > Thank you very much for the comprehensive responses. The rebuttal has addressed all major concerns and clarified the reasoning behind the selected set of visual operations and the RaPR (dynamic) mechanism.

---

> > > ### Author Response · Authors · 2025-08-05
> > >
> > > Thank you once again for the constructive guidance. These has helped improve our paper a lot.

---

### Author Response · Authors · 2025-08-01
**Summary of Reviews and Author Rebuttal**

Dear Reviewers, ACs and SACs,

We would like to express our sincere gratitude for your comprehensive and insightful feedback on our paper. We are very encouraged that the reviewers recognized the novelty of our work and the significance of the problem we are addressing. **We are pleased to find that there is a consensus on the following positive points:**

**The Significance of Pixel-Space Reasoning**: The reviewers unanimously agreed that the concept of pixel-space reasoning is a valuable contribution to the field.

* Reviewer 1: "The concept of pixel-space reasoning is a significant and timely contribution that opens new directions in VLM reasoning."
* Reviewer 2: Showed strong support with a high score, reflecting the perceived importance of the work.
* Reviewer 3: The paper "proposes an interesting idea" and acknowledges that "the idea of pixel-level reasoning via a set of image operations is reasonable."
* Reviewer 4: "The core idea of pixel-space reasoning is novel and addresses a clear gap in current VLMs."

**Methodological Rigor**: The reviewers acknowledged the strength and thoughtfulness of our approach.

* Reviewer 1: "The paper is methodologically rigorous."
* Reviewer 2: found our approach "novel and effective solutions" and "very informative to other researchers", acknowledging "the mathematical intuition behind them."
* Reviewer 3: "The approach is concise and well-motivated."

**Empirical Effectiveness**: The reviewers highlighted the strength of our empirical evidence and our comprehensive experiments.

* Reviewer 1: "The experiments are comprehensive... The demonstrated performance, especially surpassing most of the open-source and proprietary models, indicates that the proposed methods meaningfully improve practical VLM capabilities."
* Reviewer 2: The model achieves "SOTA performance on a wide range of challenging tasks," and the "extensive ablation experiments show the effectiveness of each step in the training recipe."

**Clarity and Presentation**: We are glad the reviewers found the paper to be well-written and easy to understand.

* Reviewer 2: "The paper is well-written and clearly shows the importance of these tricks and the mathematical intuition behind them."
* Reviewer 4: "The paper is well-written and structured, making it easy to follow the problem formulation, methodology, and results."

We deeply appreciate these positive comments and are grateful for the reviewers' recognition of our work.

We have carefully considered all the constructive feedback and have addressed each reviewer's questions individually. **Below we plan the concrete revisions we will make to the manuscript to avoid the identified confusions and misunderstanding:**
- Improve the clarity of proposed methodology
  - the dynamic property of curiosity bonus in L207 (Reviewer wysE).
  - a discussion on developing more sophisticated intrinsic mechanisms for pixel-space reasoning to our conclusion (Reviewer wysE).
  - a dedicated section in appendix on extending visual operations to new downstream tasks(Reviewer wysE, 1tA7).
  - advantages of using RL for pixel-space reasoning (Reviewer RbYh).
  - rephrase L83 (Reviewer wysE)
  - Explicitly frame Section 5.2 as investigation into the "learning trap" (Reviewer RbYh)
  - Rephrase the argument of self-correction data in L302 (Reviewer RbYh)
- New empirical results and analysis
  - dedicated section of qualitative analysis (Reviewer wysE): We will move the qualitative examples from the appendix to the experiment section.
  - efficiency analysis of PixelReasoner in the appendix (Reviewer 1tA7)
  - additional evaluations for chart-based QA including ChartQAPro (Reviewer RbYh)
- Corrections
  - rename "visual guided search" as "SEAL" in related work L355.
  - Correct the error in Table 1 (Reviewer 1tA7)
  - revise Figures 3 and 8** for improved readability (Reviewer wysE).

We are confident that these revisions will address the reviewers' valuable feedback and significantly strengthen the final version of our paper. Thank you once again for your time and constructive guidance.

---

> ### Author Response · Authors · 2025-08-04
> **New Evaluation Results on ChartQAPro**
>
> To further evaluate our Pixel Reasoner's OOD generalization capabilities on broader domains, we have conducted a new experiment on ChartQAPro. We list our results as follows:
>
> We compare PixelReasoner-7B against its base model (Qwen-VL2.5-7B), and other leading models reported in the ChartQAPro paper. For PixelReasoner-7B, the values in parentheses next to each score indicate the rate of its pixel-space reasoning.
> | Model                | Factoid       | MCQ            | Convers.       | FactChk.       | Hypoth.       | Overall            |
> |----------------------|---------------|----------------|----------------|----------------|---------------|--------------------|
> | GPT4o                |          37.4 |          61.68 |          33.93 |          57.37 |         30.83 |              41.68 |
> | Gemini-Flash-2.0     |         51.51 |          69.15 |          43.84 |          67.62 |         39.89 |              53.66 |
> | Gemini-Flash-1.5     |         42.37 |          64.01 |          40.17 |          56.14 |         39.42 |              45.97 |
> | Claude Sonnet 3.5    |         53.61 |          78.03 |          43.84 |          65.16 |         46.11 |          **55.81** |
> | Intern-VL2.5-1B      |          5.45 |           0.46 |          14.86 |          21.17 |         17.08 |               8.96 |
> | Intern-VL2.5-2B      |          9.42 |           6.07 |          13.02 |          36.06 |         19.23 |              13.46 |
> | DeepSeek-VL2-3.4B    |          9.63 |            1.4 |          18.09 |          38.11 |         23.25 |              14.33 |
> | Phi 3.5-Vision-4B    |         10.55 |          32.71 |           27.2 |           8.19 |          8.16 |              15.23 |
> | Qwen-VL2-7B          |         32.95 |          46.26 |           37.6 |           50.4 |         29.65 |              37.17 |
> | Intern-VL2.5-8B      |         29.53 |          23.36 |          28.87 |          56.14 |         27.73 |              31.99 |
> | LLaMA 3.2-Vision-11B |         19.65 |          47.66 |          19.15 |          44.45 |          13.1 |              25.43 |
> | Qwen-VL2.5-7B        |         37.28 |          66.12 |          36.47 |          58.19 |         45.82 |              43.27 |
> | PixelReasoner-7B     | 40.82 (14.2%) | 67.75 (11.21%) | 44.45 (19.74%) | 67.62 (10.24%) | 45.92 (10.2%) | **47.97 (15.02%)** |
>
> Our analysis highlights the following key findings:
>
> - **State-of-the-Art Open-Source Performance:** PixelReasoner-7B establishes a new state-of-the-art among open-source models with an overall score of 47.97. It outperforms its base model, Qwen-VL2.5-7B, by 4.7 absolute points on the overall score.
>
> - **Generalization:** This strong performance is achieved without any fine-tuning on chart-related datasets. This result underscores the excellent generalization capability of our model's inherent pixel-space reasoning mechanism for a complex task it was not explicitly trained for.
>
> - **Computational Efficiency:** These performance gains are realized without introducing significant computational overhead. The overall RaPR is 15%, making our approach both effective and highly efficient compared to the base model.
>
> - **Competitive with Proprietary Models:** While Claude 3.5 Sonnet leads overall, our open-source model is highly competitive, surpassing other powerful proprietary models like GPT-4o (41.68) and Gemini 1.5-Flash (45.97).
>
> - **Limitation on Unanswerable Questions:** A detailed error analysis revealed a key limitation: PixelReasoner-7B currently fails on most unanswerable questions within the benchmark. Addressing this failure mode is a clear priority and a promising direction for future work.
>
> The evaluation on ChartQAPro has significantly strengthened our paper by demonstrating the robust generalization of PixelReasoner. We believe these new results underscore the value of our contribution, and we will integrate this analysis into the revised manuscript.

---

### Note · Authors · 2025-08-14

We sincerely thank the Reviewers, Area Chairs, and Senior Area Chairs for their time and their comprehensive and insightful feedback on our manuscript.

**We are encouraged that the reviewers reached a consensus on several key strengths of our work**, including: The Significance of Pixel-Space Reasoning, Methodological Rigor, Empirical Effectiveness and Paper Writing.

**The reviewers also raised insightful questions and provided constructive suggestions for improvement.** The primary points centered on:
-   Scope of Visual Operations and the extensibility of our approach.
-   Clarifications on our methodology, such as the the dynamic properties of the curiosity bonus and risk control of data synthesis
-   Deeper analysis into efficiency trade-offs and failure modes of pixel-space reasoning.
-   Further empirical validation on additional benchmarks like `ChartQAPro`.

**We have addressed these points individually in our rebuttals and have provided new results and analyses as requested.**

Based on this constructive feedback, **we have also outlined a concrete revision plan, as detailed in the summary below.** We are committed to incorporating all suggested changes, including adding new experiments, clarifications, and improving the overall presentation to significantly strengthen the final version of our paper.

We thank you once again for your time and constructive guidance.

Authors of the Submission-16307

---

### Decision · Program_Chairs · 2025-09-17

**Decision:**

Accept (poster)

**Comment:**

The paper proposes pixel-space reasoning for VLMs, showing that models can perform visual operations like zoom-in or select-frame as part of their reasoning, and introduces a two-stage training process combining instruction tuning with curiosity-driven RL. The main strengths highlighted by the reviewers are the novelty of the pixel-space reasoning paradigm, the strong empirical results across diverse benchmarks, the methodological rigor of combining instruction tuning with RL to overcome the learning trap, and the overall clarity and presentation. Weaknesses raised include the narrow scope of visual operations, the lack of deeper theoretical analysis of the learning trap, concerns about data quality in synthetic traces, missing baselines, and a need for more qualitative examples. Reviewers also asked for more results on benchmarks like ChartQAPro and deeper discussion on efficiency trade-offs and failure modes.

During rebuttal, the authors provided detailed clarifications on the curiosity bonus mechanism, explained the extensibility of visual operations, and committed to correcting errors in baselines and tables. They also added new experimental results on ChartQAPro showing strong open-source performance and competitive generalization against proprietary models, which addressed the main criticism of limited evaluation. Some reviewers remained cautious about the scope and dataset limitations, but they acknowledged that the responses were clear and that the additional experiments strengthened the contribution. Overall, the consensus after rebuttal is that the novelty, methodological soundness, and strong empirical evidence outweigh the weaknesses. I recommend accept.